# Mapping and Measuring the Phenomenon of Precariousness in the Labour Market: Challenges and Implications

Petros Kosmas *, Antonis Theocharous, Elias Ioakimoglou, Petros Giannoulis, Maria Panagopoulou, Hristo Andreev and Aggeliki Vatikioti

School of Management and Economics, Heraclitus Research Centre, Cyprus University of Technology, 30 Arch. Kyprianos Str., Limassol 3036, Cyprus
* Correspondence: petros.kosmas@cut.ac.cy

**Abstract:** This research article presents an empirical model that takes economic vulnerability into consideration to measure and address the phenomenon of precarious work and precariousness. In order to achieve this, three satisfactory indicators were formulated, consisting of both individual and institutional levels and taking into account the country-specific relationships among the variables, depending on country-specific conditions. Based on this, the choice of homeownership is introduced instead of the eligibility for employment benefits. In this way, precarity has been examined as a condition in which precariousness and economic vulnerability intersect and interact. In Cyprus, 9.5% of the workers in Cyprus were classified as precarious, while 4.4% were classified as being in precarity (i.e., precarious and economically vulnerable). The empirical findings revealed that precariousness was related to gender, migration, and the employment sector, which is consistent with the well-known literature. One of the most noteworthy findings was the high number of female migrant domestic workers in Cyprus. In this study, new variables and novel empirical approaches were introduced into the discussion of precarious work and precariousness, which may eventually lead to new theoretical and policy avenues for reducing or eliminating this phenomenon.

**Keywords:** labour market; precariousness; precarious work; economic vulnerability

## 1. Introduction

The concept of precarious work manifests itself in a number of diverse ways, including in-work poverty related to income insecurity (Paugam 2017; da Silva and Turrini 2015; Barbier 2004; Olsthoorn 2014; Rodgers and Rodgers 1989), as well as job insecurity related to the nature of many types of employment relations (Kalleberg 2009, 2011, 2018; Spyridakis 2018; Olsthoorn 2014). In this context, the present study explores precarious work and precariousness according to a framework that encompasses employment relationships that provide low levels of control over work, low income, and low social protection.

This study provides an empirical investigation into the rise of precarious work and the extent of precariousness in the labour market of Cyprus. In order to address this phenomenon, two indicators were constructed and tested in Cyprus. Indicator 1 examines precariousness by focusing on the labour market, and Indicator 2 examines economic vulnerability as a social reproduction process. Further, a synthetic indicator was constructed, which measures precarity as a condition of precariousness intersecting with economic vulnerability (Olsthoorn 2014; Kalleberg 2011; Pitrou 1978).

Initially, an overview of the current debates on precarious work and precariousness is provided, as well as the challenges inherent when applying the methodological approach and addressing the phenomenon empirically. As a result of using a methodological approach in the current study, answers to the research questions posed are provided. Utilising microdata from the Eurostat (2020), the individuals who meet the above-mentioned criteria of precariousness were identified, and their main characteristics are described (age, gender, level of education, work experience, job specialisation, employment relationship etc.).

Classification tree analysis (CTA) was used to describe the internal structure of precarious work and identify the factors that affect labour income and the probability of workers experiencing precariousness. Utilising linear regression analysis (LRA), the factors contributing to low wages for precarious workers are identified following the estimation of the indicator of precariousness and the identification of precarious workers. Logistic regression analysis (LoRA) was used to estimate the indicators against worker characteristics (level of education, work experience, etc.) in order to determine which factors are associated with precariousness among workers. Additionally, CTA was utilised to capture precarious workers as being in a condition of precarity.

Considering the complexity of the phenomena under study, a focus group was conducted to elaborate and investigate further research questions and fill in any possible research gaps from the previous empirical analysis.

The paper concludes with a discussion of the key findings from this investigation and their implications for future research and policy guidance.

### 1.1. Theoretical and Conceptual Framework

Scholars have become aware of the term precariousness following an intervention by the French sociologist Pierre Bourdieu in an international debate on the issue (Grenoble, 12–13 December 1997); his main argument was that labour market insecurity is prevalent both in the private and public sectors. He further argued that this was due to the increase in precarious forms of employment (temporary, part-time, casual work), where the effects are more or less the same in all industries but become particularly visible in the extreme case of jobless and unemployed individuals (Bourdieu 1998).

Castel (2016) argues that neoliberalism has led to a shift in employment relations characterised by the deregulation of the labour market, the reshaping of any protective schemes, and the development of disconnection (Figure 1). In a figurative sense, the integration zone is characterised by stable social relations and stable employment in the core. An outer shell is characterised by a lack of involvement in productive activities and considerable social isolation. An outer shell is characterised by exclusion or disaffiliation. The precariousness zone is at the middle of this spectrum (Castel 2003).

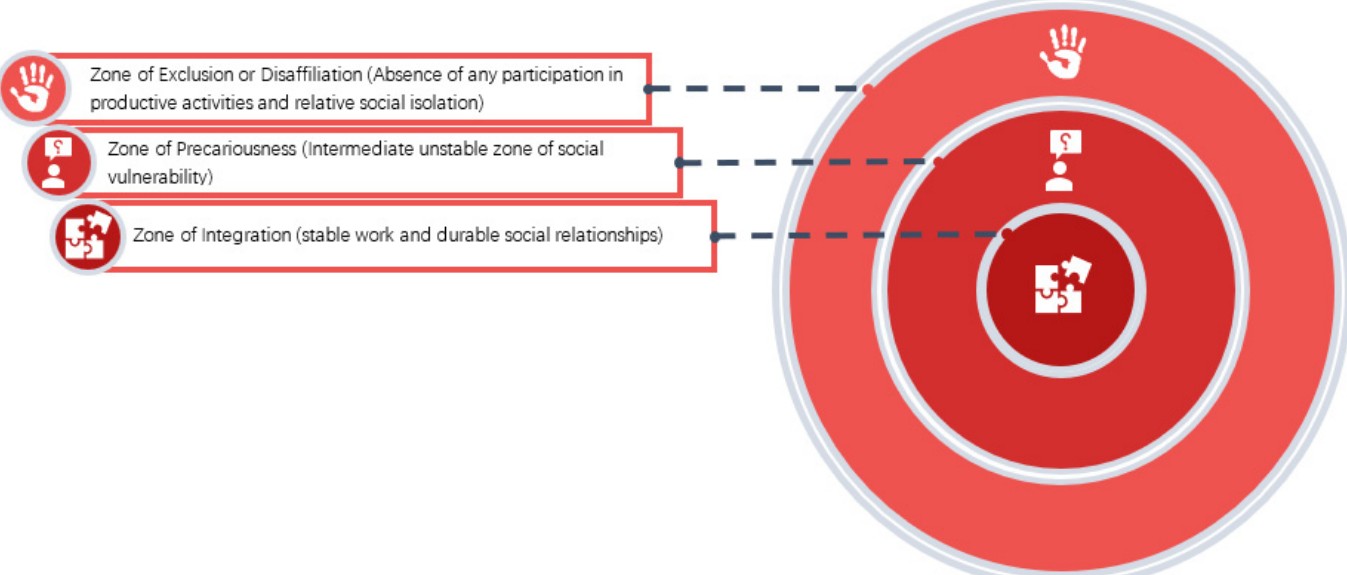

**Figure 1.** The three zones of social cohesion, according to Castel. Source: Castel (2003).

Pierre Bourdieu and Agnes Pitrou were the first to use the terms "precariousness" and "precarity" in academic discourse in 1963 and 1978, respectively (Waite 2009; Barbier 2004). However, Pitrou (1978) was the first scholar to identify factors contributing to precarious-

ness. Seven characteristics were observed by her: "*precariousness*" (which entails difficult working conditions and low wages, as well as the absence of any career prospects); "scarce as well as irregular financial resources"; "instable or unsatisfactory housing conditions"; "health problems"; "uncertainty about the future number of children"; "relative lack of social links", and a "rather precarious balance in terms of the life of the couple" (Pitrou 1978, pp. 51–64).

Standing (2009) describes precarity as a unique concept. "*To obtain by prayer*" is the Latin etymological root of precariousness. The 'precariat' is a neologism in sociology and economics for a social class formed by people suffering from precarity (Standing 2011). The members of the precariat are losing their citizenship rights: social, civil, economic, cultural, and political rights. As a consequence, they feel like supplicants and are treated as such. The continuing interaction of neoliberal policies in labour markets with restrictive immigration policies has led to the development of the "framework of hyper-precarity trap" (Lewis et al. 2015). According to De Genova (2002), there are three ways in which hyper-precarity, as a nexus of precarious employment and immigration precarity, can manifest: (a) the "displacement of daily life", which functions as the sole and ultimate mechanism of discipline; (b) a high probability of occupational injuries and deaths due to high-risk jobs in specific economic sectors (such as construction, agriculture, catering, cleaning, etc.) accompanied by limited or no access to health care, and (c) informal human networks that usually fill this gap (De Genova 2002).

The concept of precarious employment is explicated by Olsthoorn (2014) as a defining characteristic of the employment relationship, i.e., insecure jobs held by vulnerable employees who have limited entitlements to income support when unemployed (Figure 2). In his words, " . . . *precarious employment refers to employment relations that are precarious for the employee, while precarious employees and 'the precariously employed' refer to employees in an employment relation that is precarious for them . . .* " (Olsthoorn 2014, p. 424).

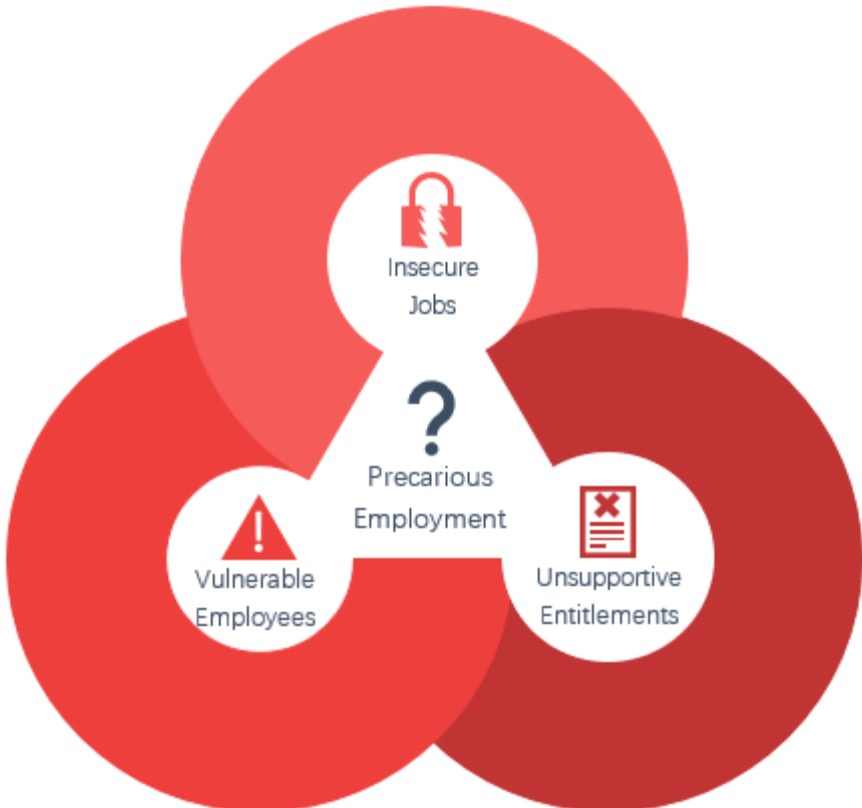

**Figure 2.** Olsthoorn's conceptualization of precarious employment. Source: Olsthoorn (2014, p. 426).

*1.2. Defining a Framework of Indicators and Contributions*

In spite of its importance, precarious employment and precariousness remain elusive and hard to measure empirically. In fact, most measurement attempts rely on nonconsolidated indicators and mediators, which raises important questions about the validity of the findings, as Olsthoorn (2014) points out. An explanation for this can be provided by the fact that there was a clear preference for indicators that only consider the type of employment relationship, i.e., whether the individual was employed on an informal basis or not. As a general concept, precarious employment is understood as a condition of threatening insecurity or risk (Olsthoorn 2014; Kalleberg 2011; Vosko 2006; Barbier 2004; Rodgers and Rodgers 1989).

Kalleberg's study empirically captures the phenomenon of precarious employment (Kalleberg 2011). He examined the rise of nonstandard employment relationships as evidence of increases in precarious employment, and, at other times, redundancies, increased unintentional job losses, long-term unemployment, weakening internal labour, and an increasing perception of precariousness. Kalleberg's notion of precariousness as job insecurity is implicit in his work, and it is associated with the use of nonstandard employment relationships as an indicator of precarious employment because nonstandard contracts can serve as proxy measures for dismissal risks.

Olsthoorn's work is of fundamental importance since it formulates two more valid and reliable indicators for measuring precarious employment (Olsthoorn 2014). In accordance with the work by Kalleberg (2011), he developed his first indicator for measuring income insecurity by using variables such as wage, supplementary income, and unemployment benefits entitlements. Here wage was considered a job-level dimension, supplementary income and an individual dimension and unemployment benefits were considered an institutional dimension (p. 427). Based on research conducted by other scholars (Barbier 2004; Rodgers and Rodgers 1989), his second indicator focused on job insecurity in relation to contract type and unemployment duration. In this case, 'the contract is considered a job-level dimension, whereas the duration of unemployment after dismissal is an individual-level dimension' (Olsthoorn 2014, p. 427). Olsthoorn's main concern regarding Indicator 2 is not whether employees can sustain themselves but whether and to what extent they are insecure regarding their employment and the severity of the consequences of job loss. Further, the two indicators were integrated to verify the coexistence of job and income insecurity at the individual level, thus allowing for a more thorough examination of precarious employment (Figure 2).

In the present study, precariousness is not only seen as a matter of employment but also of status to denote a mode of living within the 'world of work' (Kalleberg 2009, 2011, 2018; Olsthoorn 2014; Waite 2009; Barbier 2004; Pitrou 1978). Accordingly, Olsthoorn's methodology is more consistent with this concept of precariousness. As a result, it has been chosen as the basis for the methodology utilised in the current empirical study by incorporating a synthetic indicator, which takes into account Olsthoorn's five primary elements of precariousness: wage, contract type, duration of unemployment, supplementary income, and worker's entitled unemployment benefits.

The present study, however, rearranges Olsthoorn's five aspects of precariousness into two indicators; with regard to Indicator 1, it focuses on the labour market (precariousness), and Indicator 2 examines economic vulnerability (social reproduction). The reasons for this rearrangement are explained in Section 2. Then, a third indicator (Indicator 3) is calculated, which measures precarity as a condition of precariousness intersecting with economic vulnerability.

## 2. Materials and Methods

*2.1. Research Questions*

In order to measure and address the phenomenon of precarious work, an empirical approach is employed. This integrated approach aimed at providing empirically grounded results to address the following three (3) main research questions (*RQ*) under investigation.

$RQ_1$: What is the profile of precarious workers, and how can they be mapped in the Cyprus labour market?

$RQ_2$: What are the determining factors affecting the risk of precariousness?

$RQ_3$: Do economic vulnerability and precarious work meet with each other, and if so, where?

Olsthoorn's methodological conceptualisation of precarious employment is used as the basis of the empirical, analytical framework. The methodology proposed by Olsthoorn (2014) incorporates a composite indicator that takes into account the following five primary elements of precariousness:

(1)  $W_i$: the wage of $i$ worker, relative to the median;
(2)  $NpC_i$: the kind of contract (permanent or temporary work) of the $i$ worker;
(3)  $Tu_i$: the duration of unemployment periods of the $i$ worker;
(4)  $S_i$: the supplementary income of the $i$ worker (Duclos and Mercader-Prats 2005);
(5)  $UB_i$: the $i$ worker's entitled unemployment benefits ($UB$).

A synthetic indicator that integrates Olsthoorn's five aspects of precariousness is estimated for each worker. Additionally, we rearrange Olsthoorn's five aspects of precariousness into two indicators: one that focuses on the labour market and one that scrutinizes the social reproduction process. Rather than using eligibility for unemployment benefits, which is low in Cyprus, home ownership ($Ho_i$) is being used, defined as being the outright owner of her or his principal residence with no outstanding loans or mortgages, which is a rather frequent occurrence in Cyprus. According to the European Union Statistics on Income and Living Conditions-Eurostat (2020), 68.6% of the respondents were homeowners (69.7% in the EU), while 31.4% were renters (30.3% in the EU).

In addition, the economic structure, which is the system of economic relationships that remains constant in all developed countries, exhibits infinite variations because contingency circumstances differ from country to country (and, therefore, can only be evaluated empirically). According to the proposed model for analysing precarity, all variables ($W_i$, $NpC_i$, $Tu_i$, $S_i$, $Ho_i$) pertain to country-invariable relationships, whereas the thresholds describe contingent country-specific conditions framed by history, geography, productivity, and wealth. In contrast, the standard models establish operational and statistical convenience but at the cost of economically arbitrary thresholds that render blind spots concerning the specific conditions of every country.

### 2.1.1. Determining the Factors Affecting the Risk of Precariousness

The underlying objective is to describe the main characteristics of precarious workers. Having identified precarious workers in the first part of this empirical analysis, the main characteristics of precarious workers in terms of age, gender, education, professional experience, industry sector, occupations, social status, etc., are described. By using classification tree analysis (CTA), the divisional characteristics of the group of precarious workers are defined. As a next step, the factors that affect (a) the labour income of the precarious workers and (b) the likelihood of a worker becoming precarious are identified.

### 2.1.2. Intersections between Economic Vulnerability and Precariousness

The next and final step in our analysis is the examination of (a) economic vulnerability, defined as having a low income, a low propensity to save, and no outright ownership of a house and (b) precarity, as a condition in which precariousness and economic vulnerability intersect and interact.

### 2.2. Three Indicators

To empirically address the research questions under investigation, three indicators were defined. As noted earlier, in the present empirical study, the term precarity refers to a combination of precariousness (Indicator 1) and vulnerability (Indicator 2), which results in a third indicator (Indicator 3) equal to the product of Indicators 1 and 2 that identifies if and how precariousness and economic vulnerability intersect and interact.

### 2.2.1. Indicator 1: Precariousness Indicator

An indicator of precariousness is considered (Indicator 1), which is composed of $W_i$ and $NpC_i$. These two variables describe the conditions under which working capacities are sold within the labour market. Tui identifies the irregular relationship between the worker and the labour market. The $W_i$ variable represents a worker's position within the wage set or his/her precarious wage; $NpC_i$ represents the contract aspect of precarious employment by identifying whether a worker has a permanent or temporary job, and $NpC_i$ represents the length of the unemployment period for each precarious worker.

$$Precariousness\ Indicator_i = W_i * NpC_i * Tu_i \tag{1}$$

where:

*Precariousness Indicator$_i$*: is the Precariousness indicator of the $i$ worker;

$W_i$:     the wage of the $i$ worker, which is relative to the median in order to capture the worker's position in the wage set, or else, if she/he has a precarious wage;

$NpC_i$: the contract aspect of precarious employment of the $i$ worker by identifying if an individual has permanent or temporary work, which corresponds to a contract with a limited duration (value 1 for temporary work, otherwise 0);

$Tu_i$:     the duration of the unemployment periods for $i$ precarious worker. If the unemployment duration is below a certain threshold, the variable takes on value 0, and value 1 otherwise.

### 2.2.2. Indicator 2: Vulnerability

This indicator (Vulnerability Indicator) is comprised of two variables in Olsthoorn's methodology of $S_i$ and $Ho_i$ (instead of $UB_i$), both of which assess social reproduction, i.e., the ability of a worker to maintain and reproduce a working capacity on a daily basis (and perhaps on a generational basis) (Olsthoorn 2014). We consider home ownership ($Ho_i$, defined as the outright ownership of a primary residence without a mortgage or loan on it), which is quite widespread in Cyprus, rather than eligible unemployment benefits, which are extremely low. In this stage, two variables will be used to assess economic vulnerability. The first variable is *Low Savings*, which refers to a family's inability to save a significant amount of money from its income. Based on the assumption that the lower the household income, the lower their propensity to save, we utilize the *Equivalent Household Disposable Income* to estimate *Low Savings*, with the income of EUR 1500.00 per month as a threshold.

$$Vulnerability\ Indicator_i = S_i * Ho_i \tag{2}$$

### 2.2.3. Indicator 3: Precarity

Indicator 3 refers to the set of variables included in the Precariousness Indicator (Indicator 1), which contains the first set of variables ($W_i$, $NpC_i$, $Tu_i$) and their product ($W_i * NpC_i * Tu_i$) and is defined as the *Indicator of Vulnerability* (Indicator 2), which includes the second set of variables ($S_i$, $Ho_i$) and is defined as their product ($S_i * Ho_i$). The concept of precariousness is kept at this stage for workers with low employment incomes, job insecurity, and a high probability of unemployment (Indicator 1), and the concept of vulnerability is kept for workers without a home that they own outright (indicator 2).

As can be seen from Equation (3), we measure precarity by integrating all variables that represent country-invariant relations ($W_i$, $NpC_i$, $Tu_i$, $S_i$, $Ho_i$), while the thresholds describe contingent conditions which are related to country-specific characteristics, such as history, geography, productivity, and wealth. This is in opposition to standard models that establish operationally and statistically convenient but economically arbitrary thresholds, producing blind spots regarding the specific conditions of each country.

$$\begin{aligned} Precarity\ Indicator_i &= Precariousness\ Indicator_i \cdot Vulnerability\ Indicator_i \\ \Rightarrow Precarity\ Indicator_i &= (W_i * NpC_i * Tu_i) * (S_i * Ho_i) \end{aligned} \tag{3}$$

All the empirical findings of this study were derived from an analysis of the data from Eurostat, EU Statistics on Income and Living Conditions (Eurostat 2020).

Given the complex nature of the phenomenon under study, we decided to employ a focus group in addition to the quantitative design. The rationale behind this focus group was to allow a more rigorous and thorough analysis and to fill in blind spots. Thus, an online focus group discussion was organized not only because focus groups are well suited for exploratory studies in little-known domains (Brinkmann 2014) but because the real strength of this method is in providing insights into the sources of complex behaviours and motivations (Morgan and Krueger 1993). Three academics/researchers and two trade union representatives were present after being selected due to their considerable experience in precarious employment issues. The discussion revolved around three main axes: the characteristics of precarious workers in Cyprus, the role of the state, and the role of trade unions. The main findings of this process are presented in Section 3.5.

## 3. Results

### 3.1. Profiling and Mapping the Precarious Workers and Precariousness

Utilising microdata from the Eurostat (2020) and a set of three variables (low annual labour income, temporary work unemployment, and potential unemployment over one month), workers in Cyprus who were employed in precarious circumstances during 2019 were identified. *Low Annual Labour Income* (or low wage) was defined as an income lower than two-thirds of the median gross annual labour income of EUR 10,400.00 (or EUR 800.00 per month for 13 months). In Cyprus, the 13th salary is mandatory and must be paid in case of a collective agreement, a personal contract, or an employment agreement signed upon employment (Republic of Cyprus—Department of Labour Relations 2021). *Temporary Work* corresponded to a work contract with a limited duration. For each precarious worker, his or her potential unemployment duration was estimated regardless of whether she/he was unemployed during 2019. This was accomplished by regressing the duration of the unemployment of precarious workers (observed in 2019) against their individual characteristics.

In this study, it is assumed that a precarious worker in Cyprus who becomes unemployed will suffer severe economic hardships immediately following his or her job loss. Thus, the threshold value of 0 in the analysis was used, but occasionally a value of 1 was also used, merely for sensitivity analysis, resulting in a minimum and a maximum value for the estimated number of precarious workers.

CTA was employed to determine the profile of precarious workers in Cyprus (Figure 3). A total of 106,888 people had low annual labour income. Assuming that precarious workers face severe economic difficulties from their first day of unemployment, it was estimated that in 2019, there were 37,629 precarious workers, out of which 25,556 were female (approximately 2/3 of all precarious workers). There were 31,983 persons (85% of all precarious workers) who were classified as unskilled or semi-skilled (ISCO categories 4 to 9). Approximately 64% of the temporary workforce was classified as precarious.

Following the findings of this analysis, the economic activities and occupations, as well as the distribution of labour income of the precarious workers, were described. For 2019, the estimated number of precarious workers was between 9.5% and 7.3% of the total employees, depending on the assumption regarding the unemployment duration threshold. Female workers were one to two times more likely to be precarious.

Under the assumption that the economic circumstances of precarious workers in Cyprus tend to change dramatically within a short period of time after being laid off, it is estimated that in 2019, there were 37,629 precarious workers employed in Cyprus (10.5% of all wage earners) of whom 25,500 were women. Under the alternative assumption that a precarious worker's economic situation deteriorates abruptly after one month in unemployment, the number of precarious workers in 2019 was estimated to be 28,800, of which 20,900 were women.

The level of education of precarious workers is significantly lower than that of non-precarious workers (tertiary education accounts for 23.1% of males and 33.1% of females, respectively). Approximately 32% of precarious female workers obtained a primary education certificate compared to 9% of nonprecarious female workers.

Two interesting findings emerged. Firstly, female precarious workers were in a significantly more vulnerable position than their male counterparts, and subsequently, females constituted two-thirds of all precarious workers. In part, these findings may be due to the fact that female domestic workers, who were primarily immigrants, represented nearly 40% of all precarious workers.

In total, there were nine economic activities that employed 82.5% of precarious workers, whereas five economic activities (domestic work, education, restaurants, accommodation, and retail trade) were responsible for 70% of all precarious workers (Table 1). In the distribution of precarious workers by occupation and economic sector, 69.4% of all precarious workers belonged to five occupational groups: Teaching Professionals, Personal Services Workers, Sales, Child Care Workers and Teachers' Aides and Domestic Workers.

**Table 1.** Distribution of precarious workers in economic activities and occupations in 2019.

| Precarious Workers | Retail Trade | Accommodation | Restaurants | Education | Households | Total |
|---|---|---|---|---|---|---|
| Teaching Professionals | | | | 6.4% | | 6.4% |
| Personal Services Workers | | 4.5% | 4.7% | | | 9.2% |
| Sales | 6.5% | | | | | 6.5% |
| Child Care Workers and Teachers' Aides | | | | 6.2% | | 6.2% |
| Domestic Workers | | | | | 41.1% | 41.1% |
| Σ | 6.5% | 4.5% | 4.7% | 12.5% | 41.1% | 69.4% |

Data source: Eurostat (2020). Note: the classification of economic activities and occupations was adopted from the International Labour Office-ILO and the International Standard Classification of Occupations-ISCO 2008.

Workers in precarious positions tended to be young (Unt et al. 2021). In terms of their age distribution (Figure 4), half of them were under the age of 31, and the average age was 35 (while the corresponding figures for nonprecarious workers were 40 and 42). Figure 4 provides some insight into the young age of precarious workers in other sectors; half of them were under the age of 29, and the average age is 32 years. By dividing the same population into those who were working in households as domestic personnel and those who were working in other sectors (Figure 5), it becomes evident that they were even younger: half of them were under 29 years of age and an average age of 32 years. Precarious domestic workers were slightly older than their nonprecarious counterparts, with a median age of 35 years and an average age of 36 years.

Half of them have worked in paid jobs for more than 15 years, with an average of 18 years. Interestingly, the distribution of precarious workers by duration of professional experience (Figure 4) was quite different; half of them had been employed for less than five years, with the average being 10 years. In the distribution of all employees, the coefficient of variation was 0.7, while the coefficient of variation was 1.1 in the distribution of precarious workers, indicating a concentration of values around the average in this distribution (as can be seen in Figures 5 and 6). Figure 6 indicates the distribution of professional experience among all employees in Cyprus in 2019. The findings suggest that precarious workers accumulated professional experience at a much slower rate than nonprecarious workers (Figures 5–7).

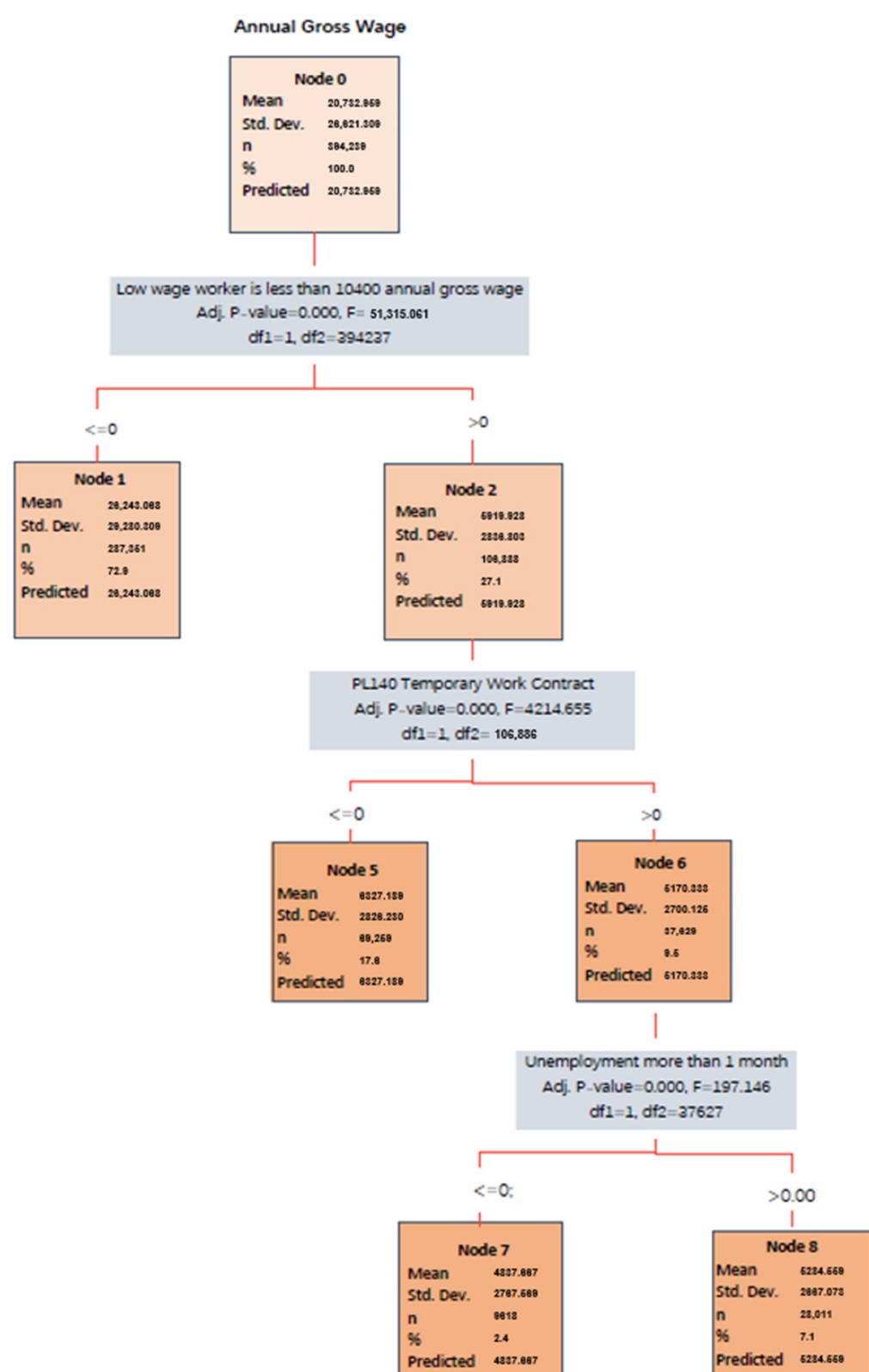

**Figure 3.** A classification tree for determining the number of precarious of workers in Cyprus and their income. Data source: Eurostat (2020).

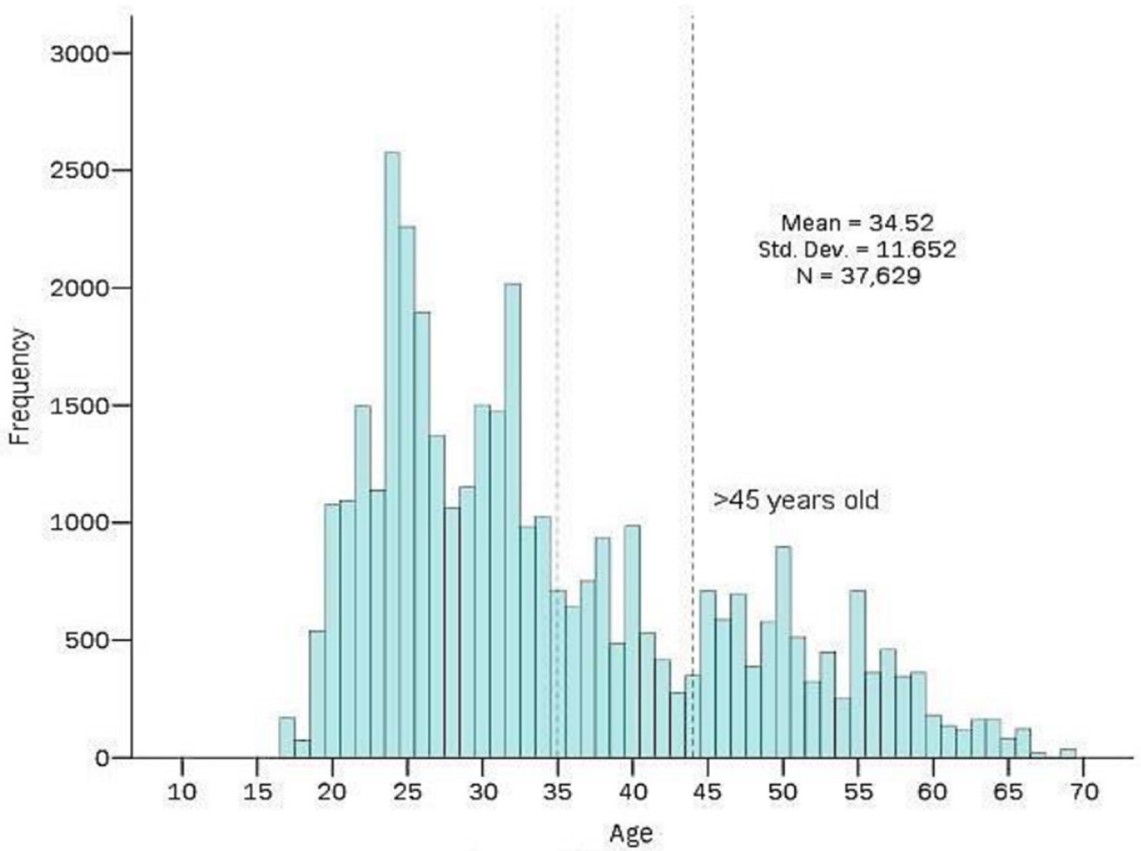

**Figure 4.** Distribution of precarious workers by age (2019). Data source: Eurostat (2020).

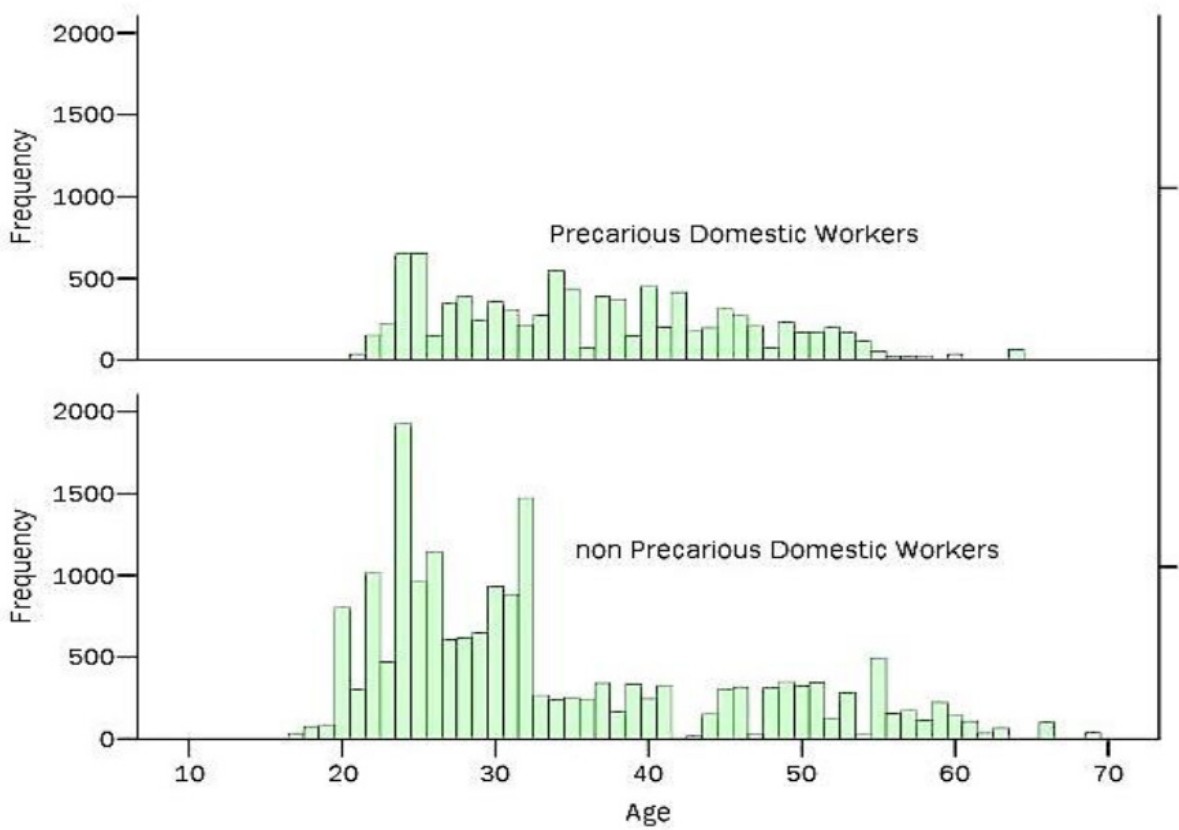

**Figure 5.** Distribution of precarious domestic workers by age (2019). Data source: Eurostat (2020).

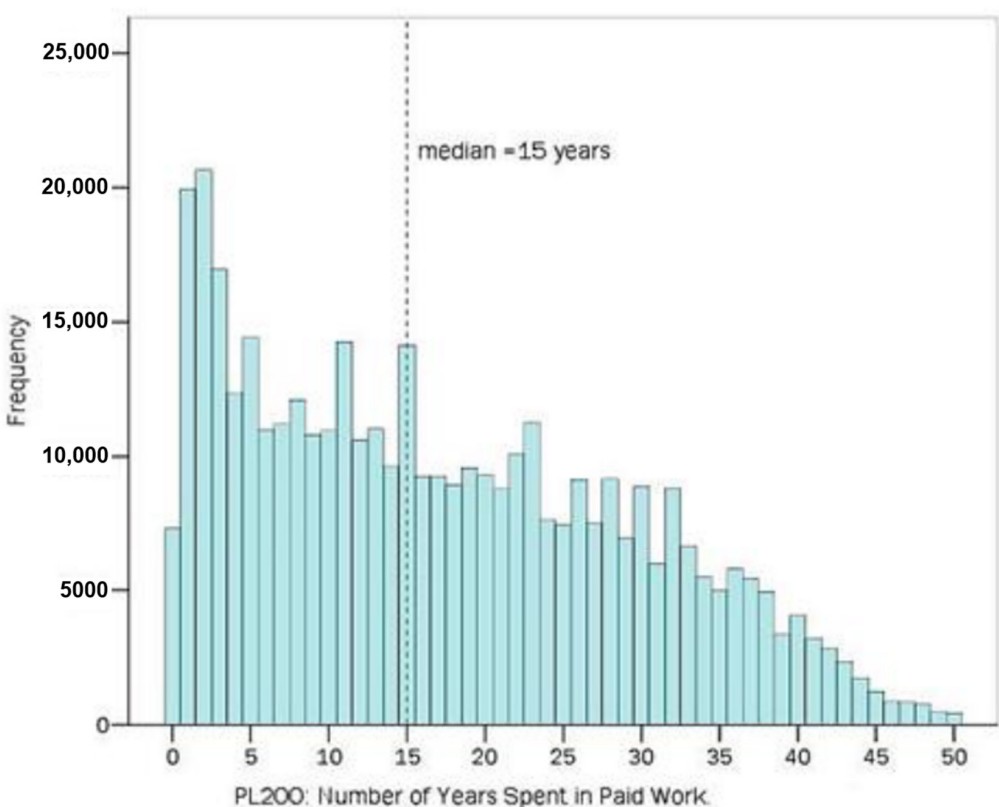

**Figure 6.** Distribution of all employees by duration of professional experience (2019). Data source: Eurostat (2020).

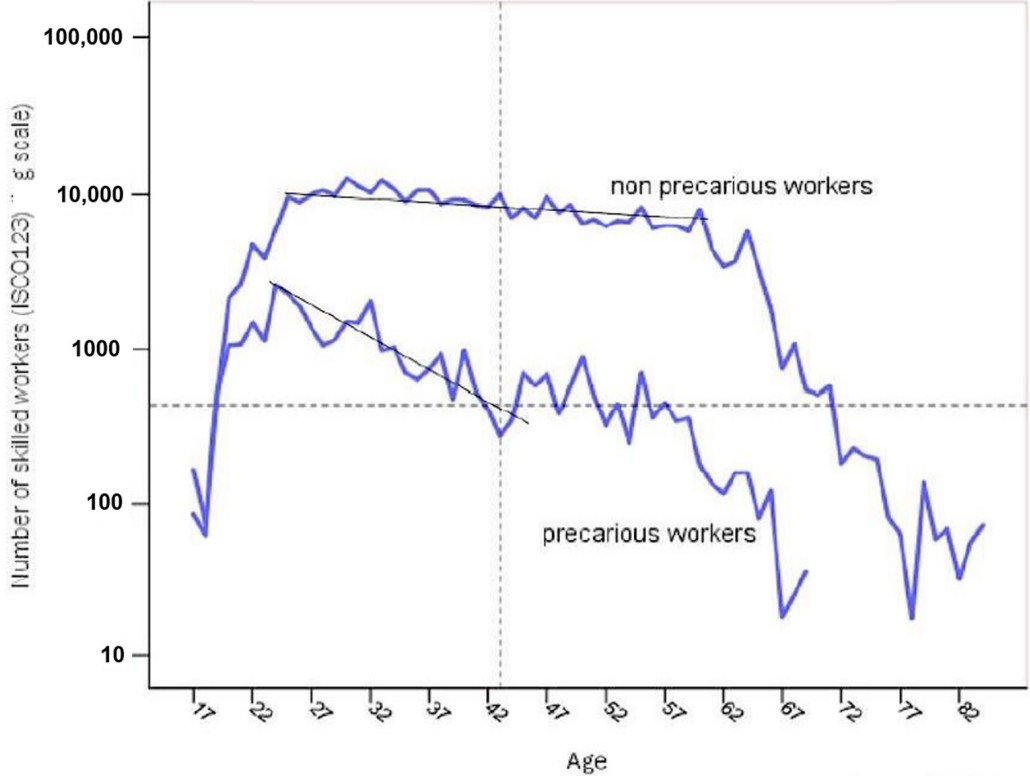

**Figure 7.** Number of skilled workers (ISCO 1, 2, 3) by age (log scale 2019). Data source: Eurostat (2020).

By using linear regression analysis (LRA) it was found that professional experience decreased with respect to four variables that appear to be tenuously related to the labour market: temporary employment, change of job since last year, part-time employment, and being unemployed for several months during the survey year.

Moreover, the number of skilled workers (ISCO 1, 2, and 3) in the population of precarious workers is lower than in the population of nonprecarious workers. For many precarious skilled workers, especially those who have completed tertiary education, precarious employment is a temporary but prolonged experience. As depicted in Figure 6, the number of skilled precarious workers decreases rapidly between the ages of 25 and 40 years. Skilled labour, however, appears to accumulate professional experience more rapidly. As a result of the factors outlined above, the accumulation of professional experience among precarious workers is low (median = 5.1 years) (Figure 8).

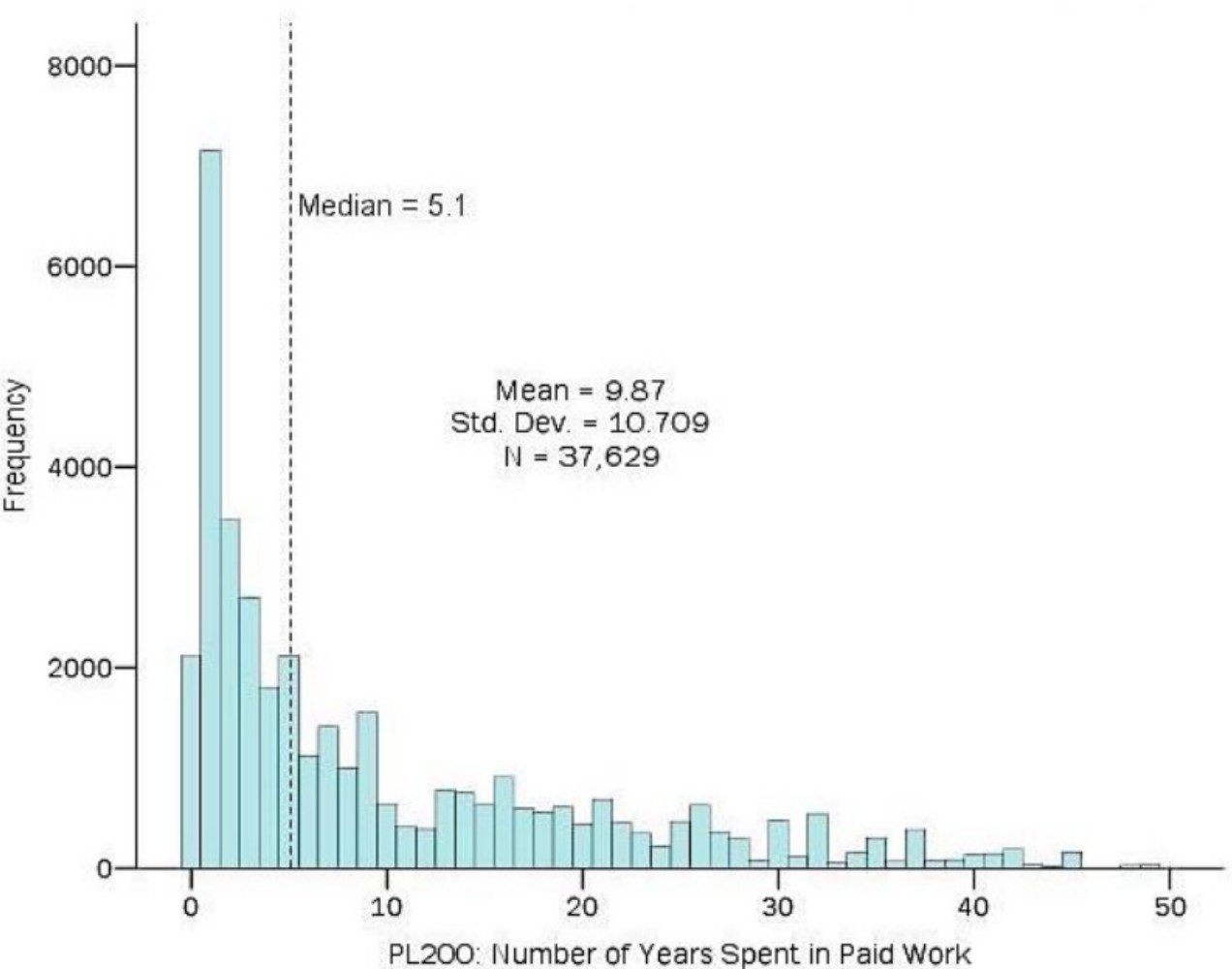

**Figure 8.** Distribution of precarious workers by professional experience. Data source: Eurostat (2020).

It can be concluded that unskilled and less skilled (ISCO 4–9) precarious workers' slow rate of professional experience accumulation is part of a vicious circle: precarious work constricts the process of accumulating professional experience, reducing, in this way, the likelihood of obtaining a better-paying, more stable, nonprecarious job. Hence, precariousness tends to reproduce itself, becoming a trap (Kretsos and Livanos 2016). Thus, the only way to break the precariousness vicious circle is through external forces, such as state labour market policies and/or the involvement of trade unions.

In order to address the issue of the overrepresentation of domestic workers within the total number of precarious workers and in order to control for potential bias, the ability to

distort the distribution of professional experience was divided. The results confirmed that the findings applied to both domestic and nondomestic precarious workers (see Figure 9).

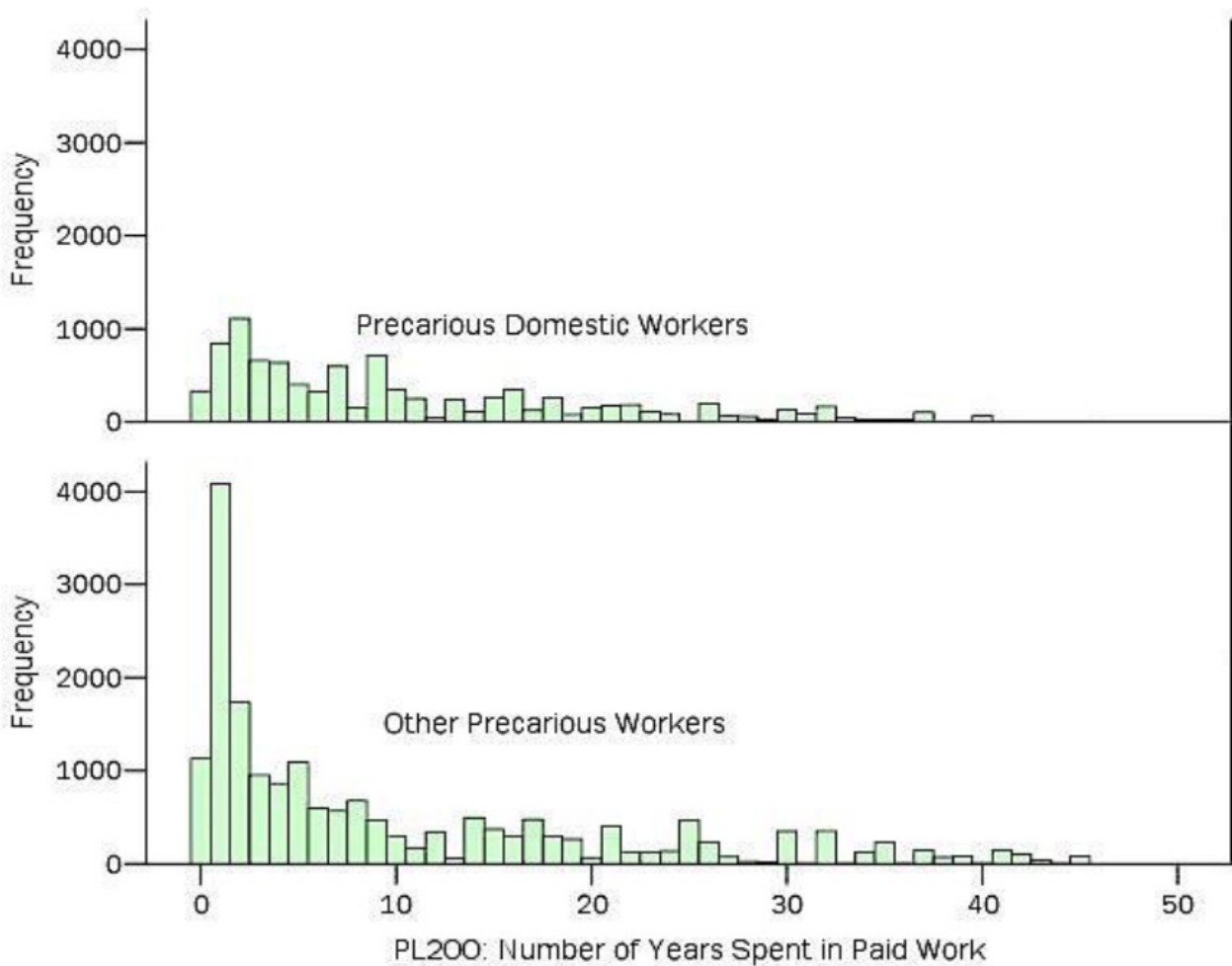

**Figure 9.** Distribution of precarious Domestic Workers by duration of professional experience (2019). Data source: Eurostat (2020).

### 3.2. Factors Affecting Labour Income of the Precarious Workers

Precarious workers' wages are determined by income, which is the annual gross earnings independent of the number of hours they work. As a proxy for annual gross wage, annual gross labour income was used (net of employer's social security contributions).

Figure 10 shows the distribution of 37,629 precarious workers by their annual gross wages in 2019. The median annual gross labour income was EUR 4980.00, and the average was EUR 5170.00. Approximately one in four precarious workers earned between EUR 7200.00 and 10,400.00 per year (or EUR 550.00 and 800.00 per month for 13 months). All remaining precarious workers earned less than EUR 7200.00 per year. The proportion of precarious workers within the income range of EUR 4000.00–5200.00 per year was high.

Figure 11 clearly illustrates the distinction between precarious domestic workers and nondomestic workers. More specifically, it shows that those in the range of EUR 4000.00–5200.00 are classified as precarious domestic workers. Based on the assumption that they were paid for 12 months, their gross monthly earnings would be in the range of EUR 350.00–450.00. However, not all domestic workers were precarious, as approximately 9528 precarious and 1500 nonprecarious domestic workers were recorded.

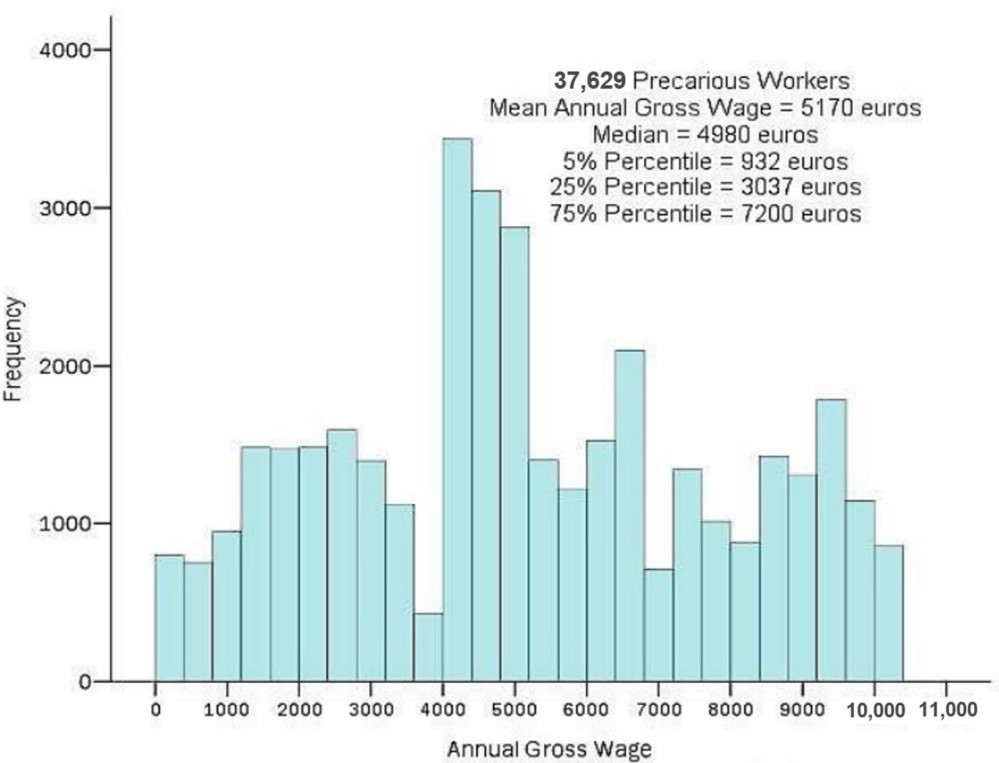

**Figure 10.** Distribution of precarious workers by annual gross labour income (2019). Data source: Eurostat (2020).

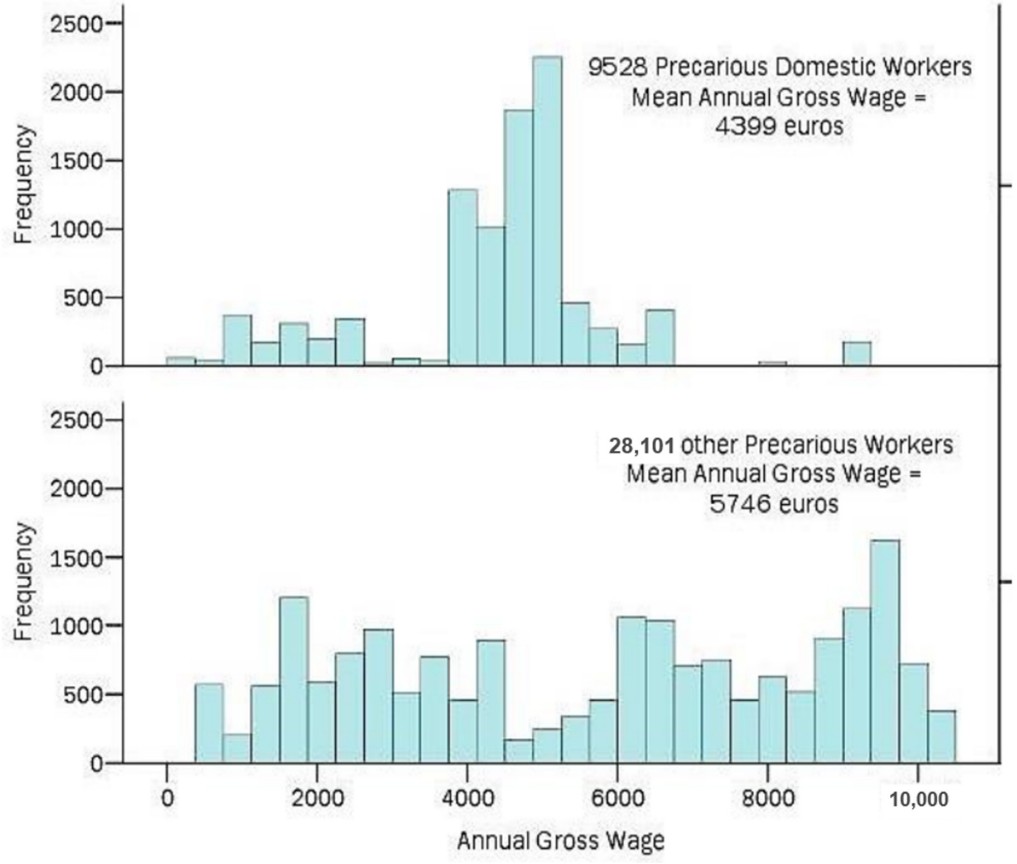

**Figure 11.** Distribution of precarious domestic workers by annual gross labour income (2019). Data source: Eurostat (2020).

If the income earned by domestic workers is excluded, the average gross labour income for nondomestic precarious employees comes to EUR 5745.00 annually (or EUR 440.00 for 13 months). Half of these workers had a gross annual income of EUR 5032.00 (or a monthly income of EUR 387.00). For 25% of nondomestic precarious workers, the average gross labour income ranged from EUR 8703.00–10,400.00 (or EUR 670.00–800.00 per month over a period of 13 months).

Nearly one in two precarious workers was unemployed for several months during 2019. The average duration was 2.75 months, whereas the median was 3.5 months. Precarious workers were more likely than nonprecarious workers to face prolonged periods of unemployment. In 2019, 19,953 out of 37,629 precarious workers (53%) were unemployed for several months before being re-employed, whereas 48,689 out of 356,610 nonprecarious workers (13.7%) were unemployed for several months before being reemployed.

Therefore, unemployment contributes substantially to the precarious low level of gross annual income and to their ongoing precariousness. As can be seen in Figure 12, precarious workers' income decreases in parallel with the duration of their unemployment. As a result of unemployment, income decreases at a constant rate of EUR 630.00 per month. It is pertinent to note that this conclusion applies to both precarious domestic and nondomestic employees (see Figure 13).

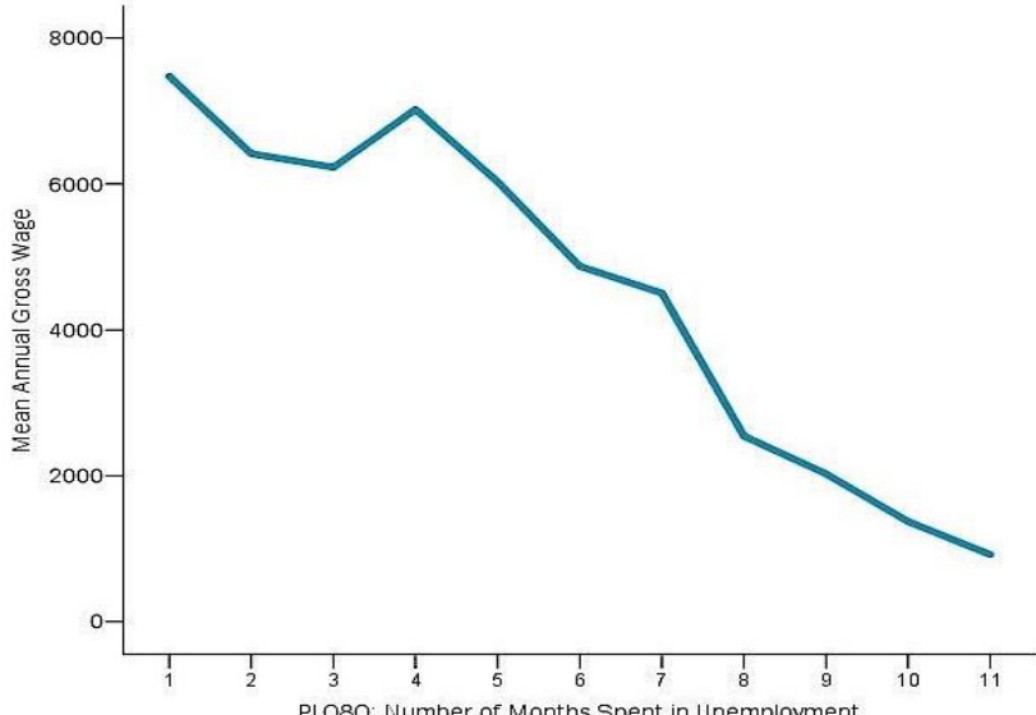

**Figure 12.** Annual gross labour income of precarious workers as a function of unemployment duration (2019). Data source: Eurostat (2020).

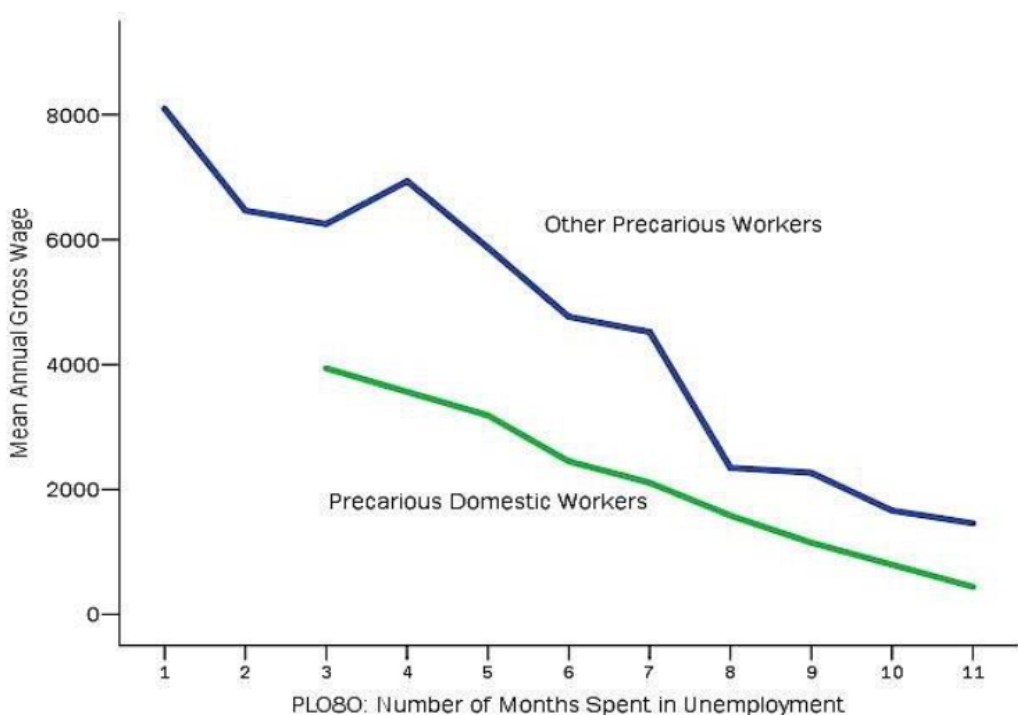

**Figure 13.** Average annual gross labour income of precarious domestic workers as a function of unemployment duration (2019). Data source: Eurostat (2020).

Utilising LRA, the factors contributing to the low wages of precarious workers were identified. These variables originate from the Eurostat (2020) (Table 2), and the results are presented in Table 3.

**Table 2.** Definition of variables of linear regression for the determination of factors affecting labour income of the precarious.

| Independent Variables | Definition | Designed to Capture |
|---|---|---|
| Female | | Employers' preferences, female employees' family responsibilities, etc. |
| ln (Age) | Age of worker | Employer's disfavour for older employees |
| Occupations 123 | Occupations in skilled intellectual work (scientific personnel, technicians etc.) | Skilled work |
| ln (Professional Experience) | Number of years spent in paid work | Professional Experience |
| Hours usually worked weekly | Hours usually worked weekly | Differential bargaining power depending on working time |
| ln (Months in Unemployment) | Unemployment duration | Lower annual income, lower bargaining power |
| Changed jobs since last year | Changed jobs since last year | Employers' preferences, lower bargaining power of the employee |
| Cypriot, Greek, OAS, NME | Citizenship | Employer's preferences and/or discrimination |
| Precarious Domestic Work | | Occupational idiosyncrasy |
| Upper Secondary Education, Tertiary Education | Education attainment level | Formal knowledge and skills, ability to perform complicated tasks |
| Dependent variable: ln (Annual Gross Labour Income) | Logarithm of Annual Gross Labour Income | |

Data source: Eurostat (2020).

**Table 3.** Results of the regression: factors affecting labour income of the precarious.

| Dependent Variable: ln (Annual Gross Labour Income) | Unstandardised Coefficients | | Stand. Coefficients | T | Sig. |
|---|---|---|---|---|---|
| | **B** | **Std Error** | | | |
| ln (Months in unemployment) | −0.586 | 0.006 | −0.494 | −93.338 | 0.000 |
| Hours usually worked weekly | 0.035 | 0.000 | 0.509 | 98.887 | 0.000 |
| Precarious domestic work | −0.769 | 0.016 | −0.261 | −48.824 | 0.000 |
| ln (Professional experience) | 0.128 | 0.003 | 0.201 | 38.124 | 0.000 |
| NME | −0.847 | 0.032 | −0.139 | −26.271 | 0.000 |
| Changed jobs since last year | −0.208 | 0.009 | −0.123 | −23.64 | 0.000 |
| Occupations 123 | 0.274 | 0.012 | 0.121 | 23.305 | 0.000 |
| Constant | 7.827 | 0.018 | | 431.656 | 0.000 |

R squared = 0.69

| Variables excluded for collinearity | Collinearity Tolerance |
|---|---|
| Female | 0.830 |
| ln Age | 0.206 |
| Cypriot | 0.663 |
| Greek | 0.658 |
| OAS | 0.115 |
| Upper Secondary Education | 0.815 |
| Tertiary Education | 0.697 |

Data source: Eurostat (2020).

Table 4 shows the variables used in the regression, and its results are presented in Table 5. It can be noted that the duration of unemployment and the number of hours typically worked each week were factors that affected the gross labour income of precarious workers. This is because unemployment duration both increased and decreased the number of hours of actual paid work in 2019. Domestic work paid significantly less than the average for other occupations. Professional experience and skilled work (ISCO 1, 2, and 3) raised wages and income, whereas changing jobs was penalized.

**Table 4.** Definition of the variables of the logistic regression for the determination of personal characteristics affecting the risk of precariousness.

| Independent Variables | Definition | Designed to Capture |
|---|---|---|
| Female | Gender (1 = Female) | Employers' preferences, female employees' family responsibilities etc |
| Education Primary, Lower, Secondary, Upper Secondary | Education attainment levels | Formal knowledge and skills, ability to perform complicated tasks |
| Professional Experience of more than 5 years | Number of years spent in paid work | Professional Experience |
| Unemployment of more than 3 months per year | Unemployment duration | Lower annual income, lower bargaining power |
| Cypriot, Greek, OAS, NME | Citizenship | Employers' preferences and/or discrimination |
| Dependent variable | Risk of the precariousness of domestic workers | Determinants of the risk of the precariousness of domestic workers |

Data source: Eurostat (2020).

**Table 5.** The results of the stepwise logistic regression: personal characteristics affecting the risk of precariousness of domestic workers.

| Dependent Variable: Risk of Precariousness of Domestic Workers (*n* = 11,054) | B | Std Error | Wald | Sig. | exp(B) |
|---|---|---|---|---|---|
| Female | 21,592 | 940,599 | 0.001 | 0.982 | $2 \times 10^{-9}$ |
| Professional Experience of more than 5 years | −2107 | 0.147 | 206,288 | 0.000 | 0.122 |
| Unemployment of more than 3 months per Year | 36,905 | 140,174 | 0.001 | 0.979 | $2 \times 10^{16}$ |
| OAS | 5176 | 0.129 | 1,614,429 | 0.000 | 176,949 |
| Constant | −22,109 | 940,599 | 0.001 | 0.981 | 0.000 |
| Nagelkerke R square = 0.658 | | | | | |
| Cox & Snell R Square = 0.363 | | | | | |
| Variables excluded by stepwise logistic regression | | | | | |
| Education Primary, Lower, Secondary, Upper Secondary | | | | | |

Data source: Eurostat (2020). Note: Total number of domestic workers 11,054 persons (199 unweighted count).

Based on the results of the preceding regression, the internal divisions of the population of precarious workers were examined. For the authors of this study, the separation of precarious workers into domestic workers and nondomestic workers is of fundamental importance since domestic workers represent slightly less than a third of precarious people and perform their duties in unusual and idiosyncratic circumstances. For this reason, the CTA examines domestic and nondomestic segments separately (Figures 14 and 15).

Figure 14 illustrates the internal divisions of precarious domestic workers. Among the 9.528 domestic workers (with no missing values), the average gross annual income is approximately EUR 4400.00. Of these individuals, 5545 had more than five years of professional experience and earned an average gross annual income of EUR 4800.00, while 3983 had less than five years of professional experience and earned an average gross annual income of EUR 3839.00. The income gap between the two groups of domestic workers was approximately 25% of the income of the less experienced.

Being unemployed for several months during the survey year is regarded as the second most significant internal divisional characteristic among domestic employees. The average reduction in income resulting from unemployment is 50%. Additionally, the third most important divisional characteristic can be attributed to the stability of employment relations. In particular, as a result of changing jobs within the last year, the average gross income was reduced by 25%.

It is noteworthy that two out of the three most important divisional characteristics refer to the stability of the employment relationship, which, in the event of a break (whether by choice to find a new job or by unemployment), is heavily affected in terms of income loss. This presumably can explain the empirical observation that domestic worker employment is less frequently interrupted by unemployment when compared to the corresponding unemployment of nondomestic workers.

Figure 15 illustrates the internal divisions within precarious employment rather than domestic employment. There are four classification variables, which are considered the most important: skilled and unskilled workers (ISCO 1, 2, 3), as well as unskilled or semi-skilled workers (ISCO 4–9), professional experience, full-time employment, and unemployment for several months during the survey year (2019). The first two classification variables are related to skills, knowledge, and the accumulation of experience, whereas the last two are associated with the continuity of an employment relationship. The division of the precarious nondomestic workers into four subgroups (see Figure 15) demonstrates that a small group of approximately 1000 (5.3%) skilled (ISCO 1, 2, 3), longer professional experience employees who earn an annual labour income of EUR 7629.00 were compared

to approximately 3200 (17.1%) skilled, shorter professional experience employees earning EUR 5391.00. With respect to unskilled or less skilled labour (ISCO 4–9), 77.6% (14,445 individuals) of those who worked full-time in 2019 earned EUR 6345.00, while those working part-time earned EUR 4704.00.

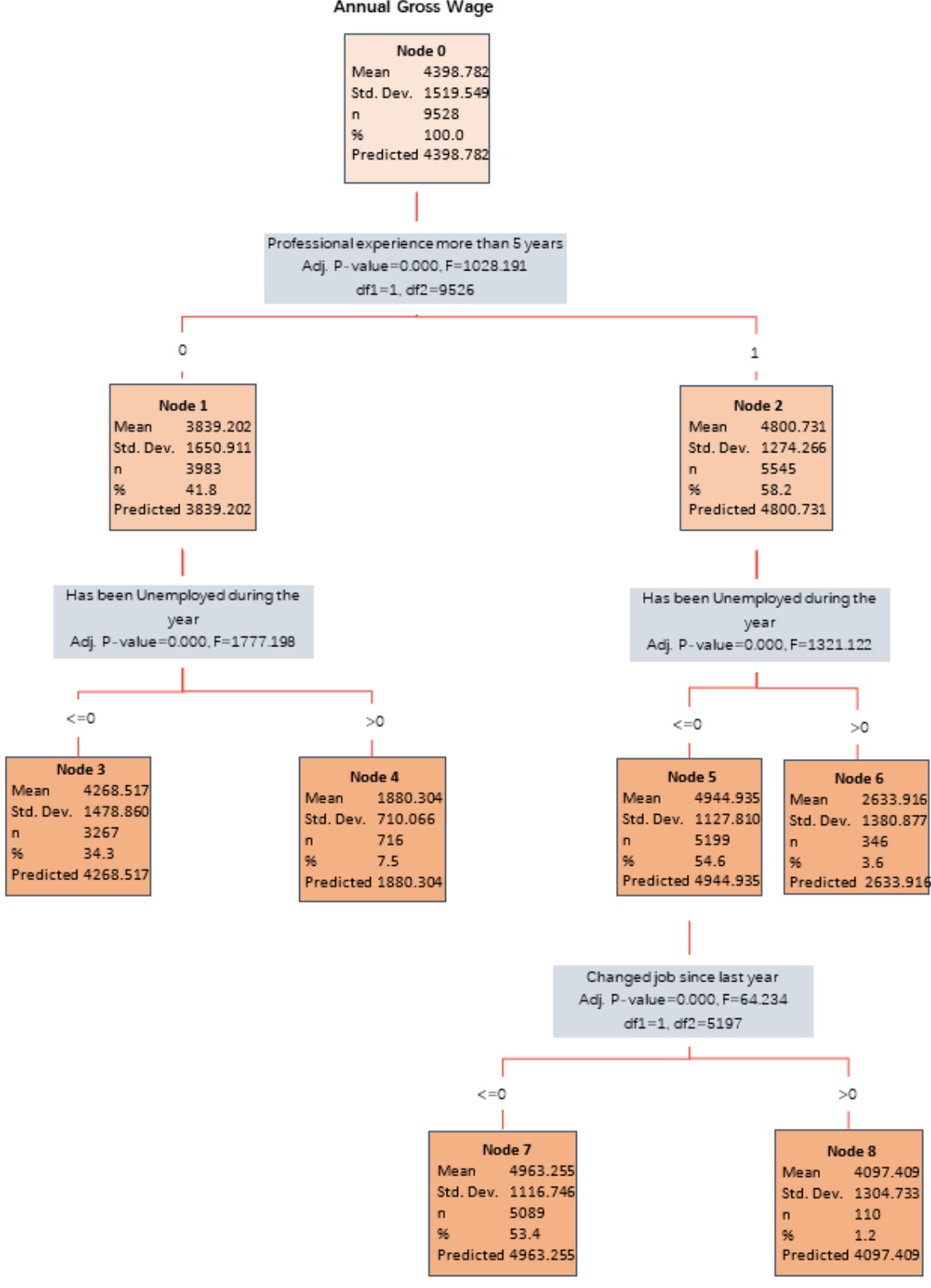

**Figure 14.** Internal divisions of precarious domestic workers. Data source: Eurostat (2020).

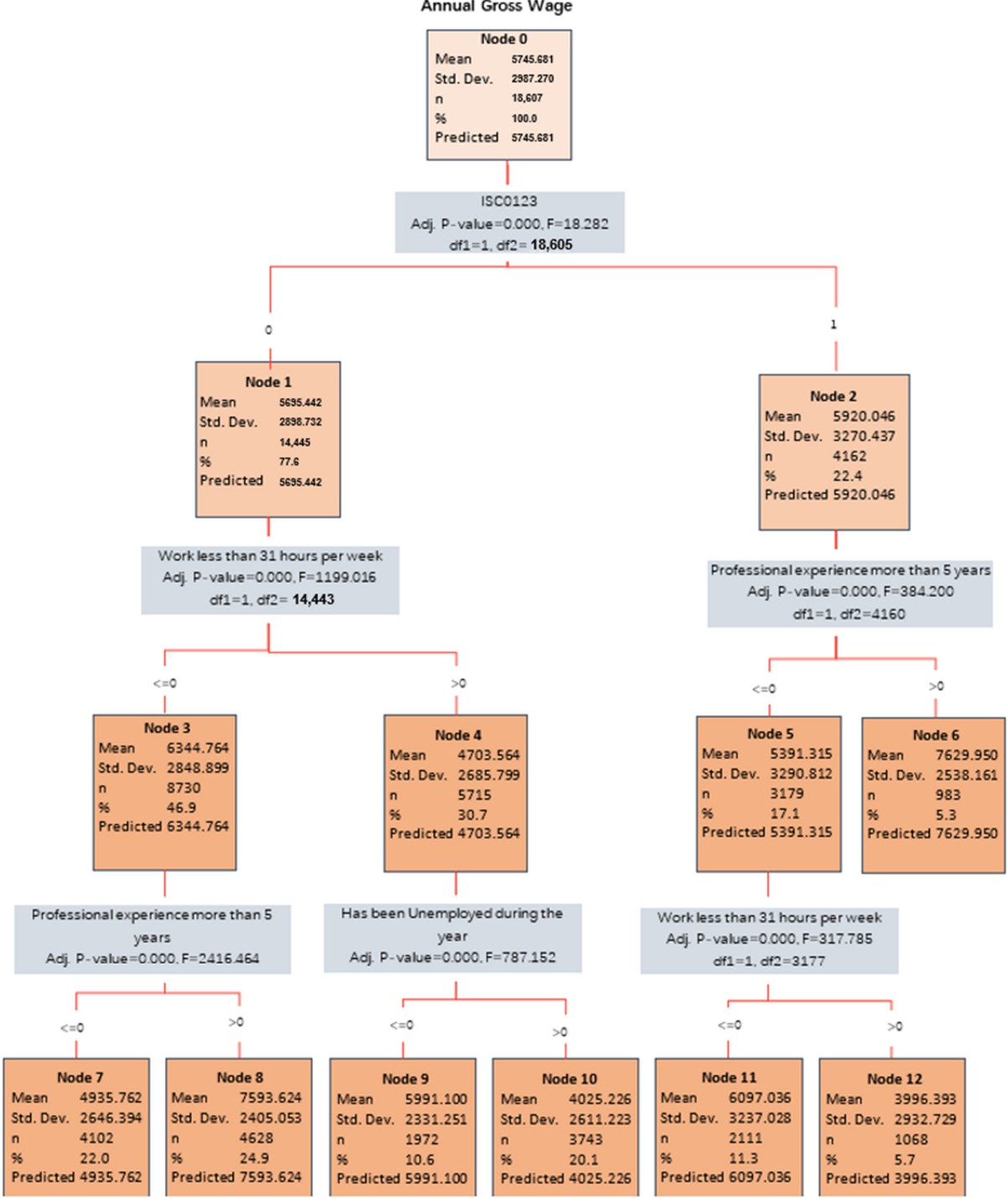

**Figure 15.** Internal divisions of precarious workers other than domestic. Data source: Eurostat (2020).

### 3.3. What Makes a Precarious Worker? Determining Factors Affecting the Risk of Precariousness

Following the estimation of the indicator of precariousness and the identification of precarious workers, logistic regression analysis was used to estimate the indicator against workers' characteristics (level of education, experience, etc.) so as to determine the factors that affect the probability of workers experiencing precariousness. Since this division is considered to be one of critical importance, two separate regression models were developed: one for domestic workers and one for nondomestic workers.

The results of the logistic regression analysis in Table 5 indicate that professional experience exceeding 5 years significantly reduce the probability of being a precarious

domestic worker. An immigrant from an Asian country had a high probability of being a precarious domestic worker. Women who were unemployed for at least three months had a higher probability of being precarious domestic workers. In other words, a young immigrant Asian woman worker with less than five years of professional experience, who had been unemployed for at least 3 months during the survey year, would most likely have worked as a domestic worker in 2019 (see Table 5).

Table 6 presents the results of the logistic regression analysis for nondomestic workers. Those with professional experience exceeding five years were less likely to become precarious workers other than domestic workers. An individual who was unemployed for at least three months during the survey year increased the likelihood mentioned above by nine times. Being a female or having completed primary or lower secondary education increased the probability by a factor of one. The probability increased by 35% if one had completed upper secondary education. Being from Asia increased the probability by six times, whereas being from the Balkans (mainly from Greece, Bulgaria, and Romania) increased the probability by a factor of one.

**Table 6.** The results of the stepwise logistic regression: personal characteristics affecting the risk of precariousness of Other than Domestic Workers.

| Dependent Variable: Risk of Precariousness of Other than Domestic Workers (*n* = 350,780) | B | Std Error | Wald | Sig. |
|---|---|---|---|---|
| Female | 0.597 | 0.017 | 1255,000 | 0.000 |
| Education Primary, Lower, Secondary | 0.700 | 0.026 | 702,787 | 0.000 |
| Upper Secondary | 0.305 | 0.018 | 275,974 | 0.000 |
| Professional Experience of more than 5 years | −1663 | 0.017 | 9422,518 | 0.000 |
| Unemployment of more than 3 months per year | 2320 | 0.018 | 15,945,344 | 0.000 |
| Greek | 0.694 | 0.033 | 432,101 | 0.000 |
| BG and RO | 0.563 | 0.029 | 376,154 | 0.000 |
| OAS | 1949 | 0.053 | 1335,857 | 0.000 |
| Constant | −2834 | 0.019 | 23,283,250 | 0.000 |
| Nagelkerke R square = 0.236 | | | | |
| Cox & Snell R Square = 0.080 | | | | |

Data source: Eurostat (2020). Note: total number of other than domestic workers are 350,780 persons (except those with missing values). Their unweighted count is 3902 persons.

### 3.4. Precarity: Where Economic Vulnerability and Precarious Work Meet and Intersect

In order to empirically address RQ3, CTA was employed. Precarious workers who were economically vulnerable in Cyprus were identified as being in a condition of precarity (see Figure 16). Of the 37,629 precarious workers, 19,304 were not outright owners of a house or apartment (approximately half of them). Most of them had a disposable household income of less than EUR 1500.00 per month (considered in our analysis as a threshold) and, consequently, were unable to save a significant amount of money to cope with the effects of unemployment (since they were not the outright owners of a residence). Accordingly, it is estimated that 17,443 workers in Cyprus were precarious and, at the same time, were economically vulnerable in 2019.

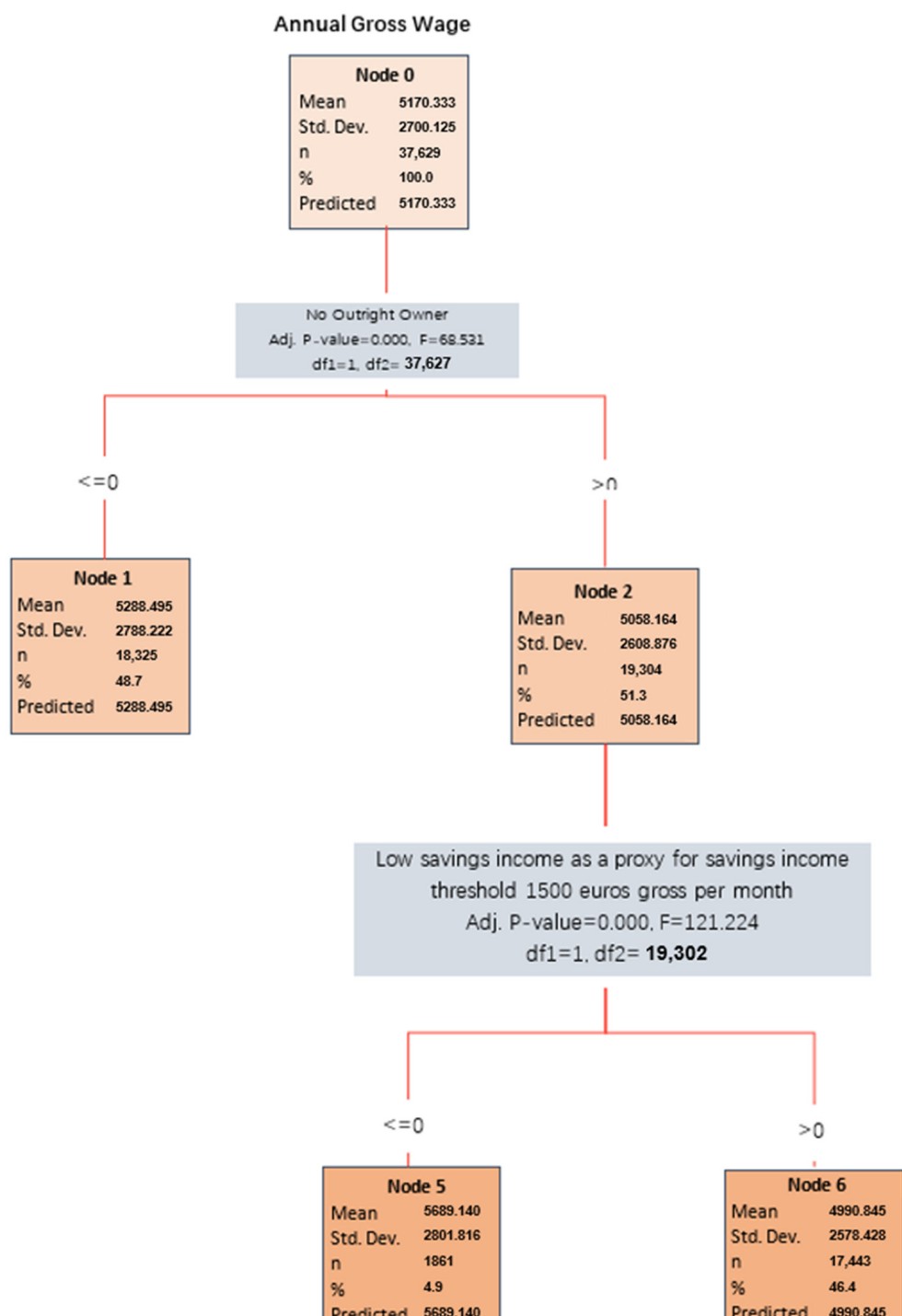

**Figure 16.** Determination of vulnerable workers in Cyprus (2019). Data source: Eurostat (2020).

Taking into consideration the results in relation to the number of workers categorised as precarious and in precarity, precarious workers constituted 9.5% of all employees in 2019, whereas those in precarity (i.e., precarious and economically vulnerable) constituted 4.4% of all employees. The percentage of workers in precarity could have been at the level of 3.3% if they were provided with the resources necessary to manage the difficulties associated with the first month of being unemployed.

Additionally, it was found that precarious work and economic vulnerability, which are the two components of precarity, were not correlated. This is outlined in a series of figures, of which the most significant ones are presented below (see Figures 17–20). The

analysis provided in previous sections of this empirical study showed that, in general, the characteristics of precarious workers and their relationship to the labour market were almost identical for vulnerable and nonvulnerable workers, with a few minor differences.

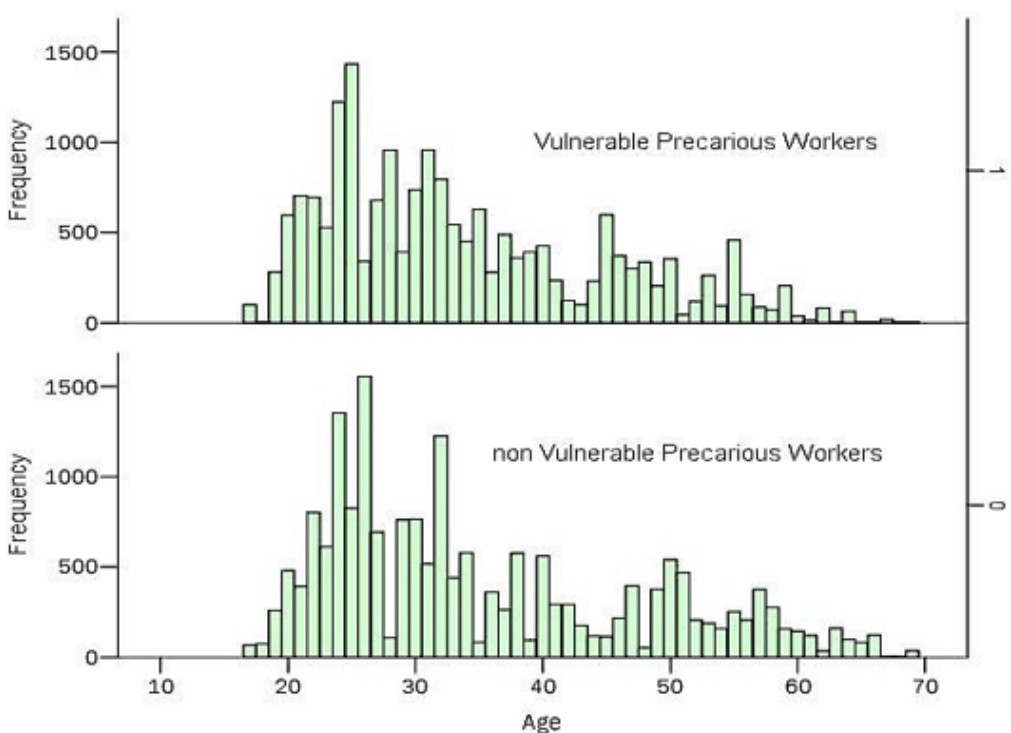

**Figure 17.** Distribution of precarious workers by age and economic vulnerability Data source: Eurostat (2020).

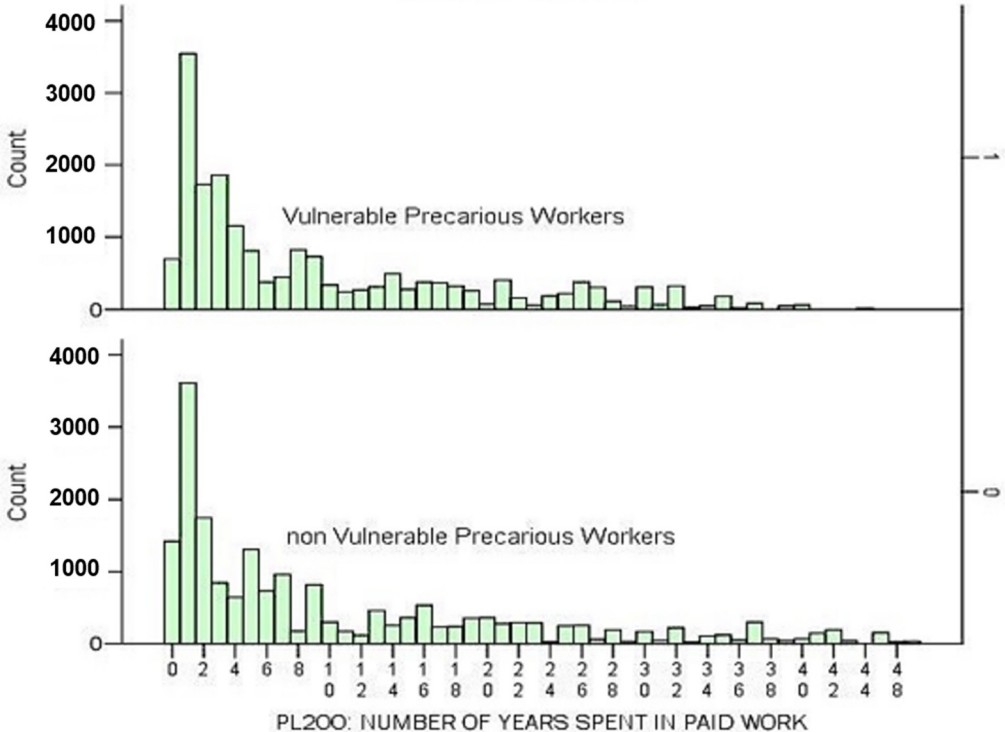

**Figure 18.** Distribution of precarious workers by professional experience and economic vulnerability (2019). Data source: Eurostat (2020).

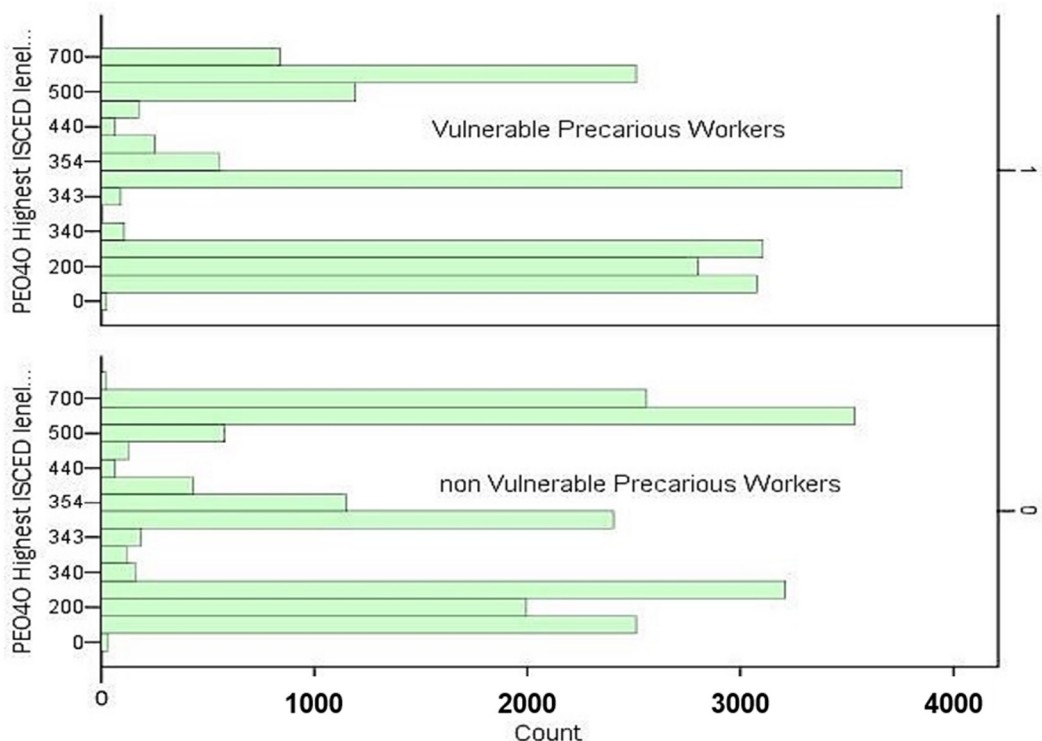

**Figure 19.** Distribution of precarious workers by education and economic vulnerability (2019). Data source: Eurostat (2020).

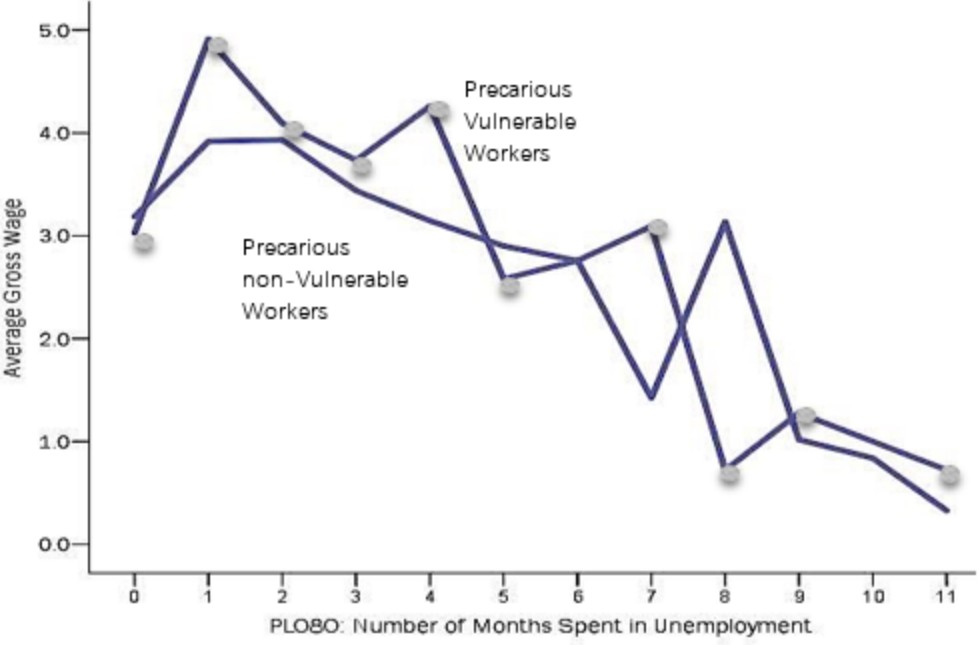

**Figure 20.** Annual gross wage as a function of duration of unemployment and economic vulnerability. Data source: Eurostat (2020).

### 3.5. Focus Group

When taking into consideration the complex nature of the phenomenon under study, it was decided to employ a qualitative analytical framework through the utilisation of a focus group. The participants of the focus group discussion perceived the concept of precariousness as a wide spectrum of people who might share some common characteristics but, at the same time, demonstrate very important differences. Thus, they argue that perhaps the

focus should be on defining the characteristics and implications of precariousness rather than strictly specifying certain social groups. Nevertheless, they agree that if the population groups included in the spectrum of precariousness are to be identified, they should include migrant workers, most self-employed workers, as well as new labour market entrants.

Regarding immigrant workers, the discussion focused on two large groups of people living and working in conditions of precariousness in Cyprus: domestic workers, the vast majority of whom are women and land workers. Despite the fact that the working conditions of domestic workers are described in their employment contract, the responsibility for ensuring compliance with the terms of the contract does not lie with the Ministry of Labour and Social Insurance but with the Police and the Migration Department (Ministry of Interior). Employers and private employment agencies are able to comply with or violate their contractual obligations without any consequences, thus setting domestic workers at a higher risk of precariousness. In addition, until 2019, domestic workers were contractually not allowed to be members of trade unions in Cyprus. Even after 2019, when this provision was deemed unconstitutional, there are no domestic workers among union members. In order to reverse these extremely unfavourable conditions for domestic workers, participants suggested that unions should try to effectively incorporate domestic workers into their ranks through targeted campaigns. Finally, a crucial issue is that the whole process of regulating domestic workers' employment—from the issuance of employment permits to the monitoring of their employment conditions—should become the sole responsibility of the Ministry of Labour, as this is the case for all other employees.

Land workers who work–and very often live–on farms also face similar issues. According to the participants, there is no monitoring of whether the terms of their employment contracts are implemented, both regarding their salary and the living conditions standards (housing, food, other resources, etc.). It was suggested that the way to deal with the issue would be to set up and operate a wide range of joint control bodies consisting of Ministry of Labour officials, trade unions, and agricultural organisations. The focus group participants also considered it crucial to elect migrant workers from the most affected sectors of labour to key positions within the trade union movement.

Regarding self-employed workers, participants note that a very large proportion of them are included in the spectrum of precariousness because they are, in fact, pseudo-selfemployed professionals being forced into this employment status by their employers to escape from their responsibilities (clearly defined salary and working hours, health care, accident insurance, access to unemployment benefits, severance pay, etc.). This status of fictitious (or false) self-employment includes a very wide range of employees from delivery workers to the employees of the wider public sector (e.g., teachers) who have the same job duties as their full-time colleagues but do not enjoy the same employment status.

In addition, genuine self-employed people live in precarious conditions, as they do not have guaranteed jobs and/or income for the next period. The phenomenon of pseudo-selfemployment has reached alarming proportions as it is often imposed not only by private employers but also in the public sector. On the other hand, a small percentage of employees who have the expectation of greater immediate earnings also accept this situation. Utilising a "carrot and stick" method, employers often try to tempt them with slightly higher earnings if they accept to work on a self-employed status. This short-term benefit is nullified by the expulsion of all the provisions entailed in an employment contract. According to the participants, there is a need to highlight the issue of the pseudo-selfemployed workers in Cyprus, but also to highlight the pitfalls of accepting a short-term personal economic benefit to the detriment of medium–long-term individual and collective interests.

According to the focus group participants, most new entrants in the labour market are employed in precarious jobs and are likely to continue being in the same situation for many years. This is particularly true in workplaces or sectors where no unions exist or they are very weak. As a result, collective claims require the simultaneous action of new entrants and unions. In other words, new employees need to take up the initiative for a

trade union formation, and at the same time, the unions need to support them decisively and effectively in the face of the most likely hostile reactions of the employers.

Focus group participants also asserted that in order to deal with the negative consequences of precariousness, all employees in Cyprus should be entitled to a legally guaranteed minimum wage and adequate employment conditions, even if a sectoral collective agreement is not in force. However, according to a participating trade unionist, the provision for a national minimum wage should only apply to employees who are not covered by sectoral collective agreements. That is, when sectoral agreements are in force, the salary provided by the contract should be provided, whereas, in cases where no such agreement exists, the national minimum wage should apply. The trade unionist made explicit reference to the fact that, with the current conditions in the labour market, it is more likely that wages will be driven down by the horizontal application of a minimum wage. The trade unionist refers to the sectoral collective agreement in the construction industry, which sets the full-time salary of semi-skilled workers at EUR 1800.00. If a horizontal minimum wage that prevails over a sectoral agreement was established, employers would only have a legal obligation to pay the minimum wage, which would definitely be significantly lower. Along with the establishment of a national minimum wage, the expansion and legal establishment of collective agreements should be claimed in as many labour sectors as possible.

## 4. Discussion and Conclusions

This article aims to contribute primarily to the theoretical discussion by proposing a more accurate method for measuring precariousness. A method that does not rely solely on work relations, but rather, it attempts to construct three indicators that examine the other defining attributes of precarious workers. In comparison to recent literature, a potential strength of this article is that it attempts to extend the existing theoretical approaches to analysing economic vulnerability, a concept that is difficult to measure due to the complexity of determining satisfactory indicators. Our empirical analysis includes individual and institutional levels and takes the country-specific relationships between the variables into consideration, depending on country-specific conditions that are too often overlooked in other standard models in the pursuit of efficiency. With respect to these issues, the article discusses the argument for replacing eligibility for work benefits with home ownership in order to advance the existing theory in this area (Kalleberg 2009, 2018; García-Pérez et al. 2017; Olsthoorn 2014; Puig-Barrachina et al. 2014; Tompa et al. 2007; Rodgers and Rodgers 1989). Essentially, this is the current work's major theoretical contribution.

The present research outlines an empirical procedure for measuring and addressing the phenomenon of precarious work and precariousness using three stages. Based on the Eurostat (2020) data for Cyprus, the characteristics of precarious employees, as well as the factors contributing to precariousness, have been identified. In "the new world of work", precarious workers comprise a heterogeneous group of individuals dealing with the multiple aspects of precariousness.

Among the precarious workers in Cyprus, women, immigrants, and young people in the labour market constituted the majority. Females constituted two-thirds of all precarious workers and were in a significantly more vulnerable position than their male counterparts. More than 40% of precarious workers were female migrant domestic workers. Over half of all precarious workers were under the age of 31.

The nature of employment relationships affects workers' employment status since many temporary employees are employed on a precarious basis. In contrast, job specialisation was a crucial factor in ensuring safe work since the participation of skilled workers in precarious employment was relatively low. Instead, precarious work was associated with low and semi-skilled work, with a concentration in five main categories of professions (ISCO categories 4 to 9). Approximately half of the precarious workers had paid work experience for less than 5 years, while the total number of employees had an average of 14 years of paid work experience. This finding showed that precarious workers were accumulating professional experience at a much slower rate than nonprecarious workers.

Therefore, precariousness appears to be a matter of gender, migration, and the employment sector that requires political will and effective policy solutions in Cyprus if the goal is to limit the phenomenon.

A key finding of this study is that precariousness tends to reproduce itself and become a downward spiral that traps workers in a precarious existence (Kretsos and Livanos 2016). The empirical findings have clearly demonstrated that the unskilled and less skilled precarious, slow rate of professional experience accumulation was part of a vicious circle. Precarious work constricted the process of accumulating professional experience, reducing the likelihood of obtaining a better-paying, more stable, nonprecarious job. Therefore, only external forces, such as government and/or trade union interventions, will be able to break this downward spiral that keeps workers in a precarious position by implementing appropriate labour market reforms.

Unemployment at regular intervals tended to be more prevalent among precarious workers compared to nonprecarious workers. Additionally, both unemployment and the number of working hours contributed significantly to the precarious' low-income levels and continued precariousness. As indicated in the study, precarious workers' income decreased at a constant rate of EUR 630.00 per month during the duration of their unemployment.

Migrant domestic workers find themselves trapped in conditions of 'hyper-precarity' with different vulnerabilities compared to those of other groups of workers (Hadjigeorgiou 2020; Lewis et al. 2015). The empirical findings showed that the event of a break, whether by choice in order to find a new job or by unemployment, was heavily penalised in terms of lost income. This can explain why domestic workers' employment was less frequently interrupted by unemployment as compared to their nondomestic counterparts, but at the same time, due to the existing legislation, their employers are able to comply with or violate their contractual obligations without any consequences, thus exposing domestic workers to a higher risk of precariousness. Overall, however, this can be explained as the result of migrant domestic workers in Cyprus experiencing the three elements which constitute the 'framework of hyper-precarity trap': precariousness, vulnerability, and legislation (Lewis et al. 2015; Anderson 2007). For nondomestic precarious workers, skills, working experience and the continuity of the employment relationship were the main contributors of income.

The geography of precariousness in Cyprus lies between a social protection zone and the zone of social exclusion (Castel 2003). Precarious work involves a constant rotation between temporary or nontypical work and periods of unemployment. Due to this situation, many workers in the Cyprus labour market do not have a regular income that would provide them with social security. This would facilitate their integration into society. As a result of underemployment, incomes were often insufficient to support a decent standard of living. There were many precarious workers whose incomes were below the official poverty threshold. Despite not being unemployed, they were covered by the working poor scheme (new poverty) (Paugam 2017, 2020).

An aspect of precarious work in Cyprus that was not uncovered in the quantitative research was revealed during the focus group. Selfemployed individuals should be included in the spectrum of precariousness as they are, in fact, false (or pseudo) selfemployment professionals that have been forced into an employment relationship by their employers. The status of false selfemployment applies to a variety of workers, from delivery workers to the employees of the broader public sector.

After identifying the characteristics of precarious employees and the factors that contribute to precariousness, our next step was to investigate precarity as a condition of precariousness intersecting with economic vulnerability. When taking into consideration the results in relation to the number of workers categorised as being in precariousness and in precarity, precarious workers constituted 9.5% of all employees in 2019, whereas those in precarity (i.e., precarious and economically vulnerable) constituted 4.4% of all employees.

In a neoliberal era, this empirical research and its implications contribute to the discussion of the phenomenon of precarious work and precariousness by (a) including

new variables and expanding the empirical approaches and (b) providing researchers, international labour organizations, governments, labour unions, employers, workers, and other stakeholders with a deeper understanding of the phenomenon, which will ultimately lead to new theoretical and policy avenues towards its reduction or even elimination.

## 5. Limitations

This study was not able to investigate the effect of the COVID-19 pandemic period on the phenomenon of precarious work and precariousness in Cyprus as a result of the fact that the relevant statistical data were unavailable or incomplete at the time of the implementation of this research (the COVID-19 pandemic period was still ongoing).

## 6. Future Research

A future study dealing with the same research topic in specific industries, such as tourism and hospitality in Cyprus, would help to identify and highlight the key characteristics and dynamics of each economic activity that contribute to precariousness and precarious employment. An additional future study suggestion that addresses the limitations of the current empirical investigation is to examine the effect of the COVID-19 -pandemic as a proxy for future crisis events on precarious work and precariousness. There is also a need to examine the issue of the pseudo selfemployed workers in Cyprus since the phenomenon of fictitious selfemployment has reached alarming proportions as it is often imposed not only by employers but also by the State.

**Author Contributions:** Conceptualization, P.K., E.I and A.T.; methodology, P.K., E.I., P.G., M.P. and A.T.; software, P.K. and E.I.; validation, P.K., P.G., A.V., A.T., M.P. and H.A.; formal analysis, A.T. and P.K.; investigation, E.I., P.K., P.G. and M.P.; resources, P.K. and A.V.; data curation, E.I., P.K., P.G. and M.P.; writing—original draft preparation, P.K.; writing—review and editing, A.T., P.G. and P.K.; visualization, H.A. and E.I.; supervision, P.K. and A.T.; project administration, P.K. and H.A.; funding acquisition, P.K. All authors have read and agreed to the published version of the manuscript.

**Funding:** This paper is the intellectual output of a research project funded by the A.G. Leventis Foundation and the Hellenic Observatory at the London School of Economics and Political Science entitled "Addressing and Measuring the Phenomenon of Precariousness in Cyprus: challenges and implications". The views expressed in this publication are those of the author and should not be attributed to the Hellenic Observatory at the London School of Economics and Political Science.

**Institutional Review Board Statement:** Not applicable.

**Informed Consent Statement:** Not applicable.

**Data Availability Statement:** The data that support the findings of this study are available from the Statistical Service of the Republic of Cyprus (CYSTAT), but restrictions apply to the availability of these data, which were used under license for the current study. Data are, however, available from the authors upon reasonable request and with permission of CYSTAT.

**Conflicts of Interest:** The authors declare no conflict of interest.

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
