# Peer review of "Mapping and Measuring the Phenomenon of Precariousness in the Labour Market: Challenges and Implications"

_socsci, doi:10.3390/socsci12020061_

Round 1

Reviewer 1 Report

This is an interesting paper, utilising a specific definition of precariousness and contributing some new empirical data. While there is some merit in the attempt to frame precariousness in numerical terms, the complexity of the phenomenon does not render itself to explanation via statistical analysis. The authors understand this and complement quantitative analysis with a focus group, but the connection between the two is weak.     

The paper will benefit from:

a) further engagement with international literature, especially the more recent, and connect the findings from Cyprus with those of other scholars abroad in the discussion.

b) cutting some of the less necessary tables and figures and expanding the qualitative and analytic sections

Some more minor points are:

a)      Need to explain the notion of the 13th salary as part of the annual salary in Cyprus

b)     On p. 32 it must be clarified that the reference to 1800 euros as starting salary in construction refers to skilled (or semi-skilled) workers

Author Response

Dear Reviewer,

First of all, I would like to thank you very much for your fruitful comments and valuable suggestions.
a) I inform you that the paper was further linked to the known literature in the discussion and conclusions.
b) I provide an explanation of the 13th salary, based upon the Cyprus law.
c) The salary of 1800 euros refers to semi-skilled workers.

In addition, I would like to inform you that I have highlighted in yellow all additions, changes, or deletions deemed appropriate after considering both reviews.

Please accept my sincere thanks once again for your review.

Yours sincerely,

the author

Reviewer 2 Report

The aim of the paper is to measure and address the phenomenon of precarious work and precariousness in Cyprus. By analysing data from the EU-SILC (2020), using the Classification Tree Analysis the characteristics of precarious employees were identified, and utilising Linear Regression Analysis and Logistic Regression Analysis the factors contributing to precariousness are estimated. Finally, the results are discussed in a focus group with three academics/researchers and two trade union representatives with a considerable experience in precarious employment issues. 

The work fit the journal scope, the main article contribution is to propose a measurement of the precariousness based not only on employment relationships, but by attempting to construct three indicators that considers other important issues characterising the life conditions of precarious people. The strengths of this article, regarding the more recent literature, is to consider economic vulnerability, a difficult concept to measure because of the complexity to find some satisfactory indicators. The empirical analysis includes individual and institutional levels, and it takes into account country-specific relationships among variables, depending on country-specific conditions that in the standard models are too often overlooked in pursuit of efficiency. Regarding these issues, the choice to use the homeownership instead of the eligibility to employment benefits, well explained in the article, is convincing in this reviewer’s opinion. 

Some results confirm the well-known literature, for example the trap of precariousness and its impact on the possibility to accumulate professional experience, or the impact of unemployment on economic vulnerability. Other results are interesting in order to assess the policies in Ciprus. In particular, some data deserve attention: 40% of precarious workers were female migrant domestic workers, over half of all precarious workers were under the age of 31. Taking into account that precarious works are associated with low and semi-skilled work and are concentrated in five main categories of professions, in Ciprus precariousness seems to be a matter of gender, migration and economic sector that is asking for political solution. In this regard, the idea to discuss the results with operators and administrators is a good one and should be expanded, organising more focus with different administrators. 

The paper has not methodological inaccuracies, but the literature review is lacking of some important references important in order to take into account the institutional differences among countries: the works of Hans-Peter Blossfeld, Michael Gebel, Sonia Bertolini, Marge Unt should be useful (for example, Unt, Marge, Gebel, Michael, Bertolini, Sonia,  Deliyanni-Kouimtzi, Vassiliki, Hofäcker, Dirk have edited “Social Exclusion of Youth in Europe. The Multifaceted Consequences of Labour Market Insecurity”, Policy Press). The conclusions are consistent with the evidence and arguments presented, but the results have to be commented not only in relation to the local situation, but also to the insights of the empirical literature, in order to contribute to the growing of knowledge about the topic of economic vulnerability and its differences among countries. If the authors improve the conclusion in this direction, showing a higher knowledge of the existing empirical literature, the manuscript can be relevant for the field.

The manuscript is clear and presented in a well-structured manner, it is scientifically sound and the experimental design is more than appropriate to test the hypothesis, the manuscript’s results reproducible based on the details given in the methods section.

The figures and images are appropriate in order to improve the understanding, but they can be reported in the appendix, leaving only the most important in the text. 

Author Response

Dear Reviewer,

Thank you for reading the article thoroughly, so your fruitful  comments are absolutely accurate. Upon reviewing the revised version, you will notice that more of your suggestions have been incorporated into the manuscript.

I am very pleased to hear that you consider the option of using home ownership over employment benefits to be persuasive. In truth, this is a first attempt, to the best of my knowledge, and I have been somewhat hesitant to discuss it further. This led me to modify both the abstract and Discussion and Conclusions by emphasizing the paper's theoretical contribution.

Additionally, I would like to inform you that I have highlighted in yellow all additions, changes, or deletions that I believe are appropriate after considering both reviews.

Please accept my sincere thanks once again for fruitful comments and valuable suggestions.

Yours sincerely,

the author